

# Many-body localization in presence of cavity mediated long-range interactions

Piotr Sierant[1*], Krzysztof Biedroń[1], Giovanna Morigi[2] and Jakub Zakrzewski[1,3]

**1** Instytut Fizyki imienia Mariana Smoluchowskiego,
Uniwersytet Jagielloński, Łojasiewicza 11, 30-348 Kraków, Poland
**2** Theoretische Physik, Universität des Saarlandes, D-66123 Saarbrücken, Germany
**3** Mark Kac Complex Systems Research Center, Jagiellonian University,
Łojasiewicza 11, 30-348 Kraków, Poland

* piotr.sierant@uj.edu.pl

## Abstract

We show that a one-dimensional Hubbard model with all-to-all coupling may exhibit many-body localization in the presence of local disorder. We numerically identify the parameter space where many-body localization occurs using exact diagonalization and finite-size scaling. The time evolution from a random initial state exhibits features consistent with the localization picture. The dynamics can be observed with quantum gases in optical cavities, localization can be revealed through the time-dependent dynamics of the light emitted by the resonator.

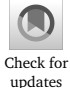
## 1  Introduction

Many-body localization (MBL) is the most robust manifestation of ergodicity breaking in interacting many-body systems. The interactions are generically considered as leading to ergodic dynamics as far as local observables are concerned. The standard formulation of this belief is the eigenvector thermalization hypothesis (ETH) [1, 2]. Numerous studies over the last decade have shown that the many-body interacting systems do not thermalize in presence of a quenched disorder. In particular, for a sufficiently strong disorder MBL may occur, leading to a long-time memory of the initial state (for recent reviews see [3] as well as a topical issue of Annalen der Physik [4]).

While most studies so far considered short-range interactions, for instance in the XXZ spin Hamiltonian (which became a paradigmatic model of MBL [5]) or bosons and fermions in optical lattices in the tight–binding limit [6–10], it is by no means clear whether MBL persists for genuinely long-range interactions, such as for Coulombic or dipolar potentials. In recent works [11] it was argued that MBL may appear in disordered long-range interacting systems, others suggest [12–15] lack of MBL for, e.g., dipole-dipole interactions in three dimensions (3D) [16, 17]. Furthermore, MBL in presence of power-law decaying tunneling elements and interactions has been studied in [18–20] where algebraic decay of correlation functions as well as of algebraic growth of entanglement entropy was found. In other studies [21–24] a model of a single spin coupled to all sites of otherwise short-range interacting systems was analysed showing that the presence of additional coupling may strongly modify MBL properties.

In the present work we numerically analyse whether MBL occurs in a disordered Hubbard model with all-to-all interactions. This model is expected to describe the dynamics of atoms in an external lattice and interacting dispersively with a mode of a standing-wave optical cavity. Such a model has been extensively studied in the past concentrating mainly, however, on ground state properties [25–37] (for a review with extensive list of references to earlier works see [38]. The certainly incomplete list of recent experimental works in that area includes [39–42]). Long-range interactions appear naturally in this system – the mode of the cavity mediates a two-body interaction whose range is as large as the system size. When the atoms are tightly confined by an external optical lattice, the cavity-mediated long-range interactions tends to order the atoms in structures maximizing the intracavity field intensity. We investigate MBL in this extended Hubbard model and with local disorder using exact diagonalization supplemented by numerical techniques for sparse Hamiltonian matrices for a gas of fermions (or bosons). Our results show features which can be attributed to the occurence of MBL in the system. We argue that these features can be revealed in the light emitted by the resonator.

The paper is structured as follows. In Sec. 2 we describe an experimentally realizable system of atoms inside a resonant cavity. We summarize the details of the derivation of the effective model in Appendix A, as described e.g. in [28, 33]. In Sec. 3 we determine the phase diagram by a finite-size scaling analysis assuming that the atoms are spinless fermions. In Sec. 4.1 we turn to dynamics of the system and provide the evidence of ergodicity breaking resulting from the interplay of disorder and interactions in the system. In Sec. 4.2 we show that

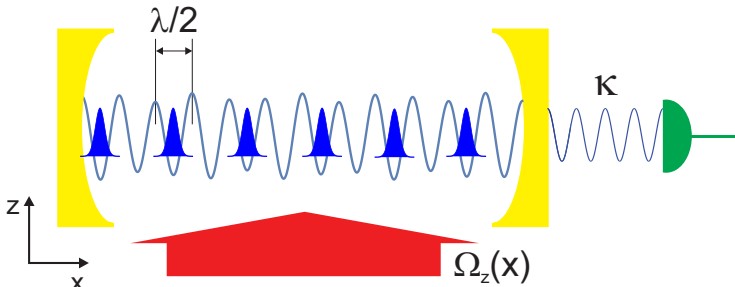

Figure 1: (color online) The Hubbard model we consider describes the dynamics of atoms in an optical lattice and interacting with the standing-wave mode of a high-finesse optical cavity. In the limit in which the atom-photon interactions are dispersive and the cavity field can be adiabatically eliminated from the atomic dynamics, the resulting Hubbard model is characterized by all-to-all interactions mediated by the cavity photons. The trasverse arrow symbolizes an external laser pumping photons into the cavity via coherent atom scattering. The additional disorder shifts the energy of the lattice potential. The quantum state of the system can be inferred by measuring the light at the cavity output, by time-of-flight measurements, or by Bragg spectroscopy using a weak probe.

the ergodicity breaking can be understood within an appropriately modified picture of local integrals of motion (LIOMs). We then argue that in cavity QED setups ergodicity breaking can be revealed via the light emitted at the cavity output. In Sec. 6 we provide arguments that the reported properties on the system are similar both for purely random disorder as well as for quasiperiodic potential. Finally, we discuss MBL for bosons loaded into the cavity system. Appendix B contains a short discussion of the nonergodicity in the specific limit of very strong cavity-mediated coupling which is outside the MBL regime; Appendix C provides the details of the time-propagation algorithm used.

## 2  Description of the system

We consider an ensemble of atoms trapped in a quasi one-dimensional geometry and tightly bound by an optical lattice. The atoms dispersively interact with an optical cavity in the regime, in which the cavity mode can be adiabatically eliminated from the dynamics. We assume that the cavity field is a perturbation to the optical lattice, so that the dynamics can be restricted to the lowest band of the external lattice and the cavity-mediated long-range interactions effectively describe all-to-all interactions between the lattice sites. The dynamics is governed by the effective Hamiltonian (for the detailed derivation see Appendix A):

$$H = H_\mathrm{A} + H_\mathrm{C}, \tag{1}$$

with $H_\mathrm{A}$ being the standard Hubbard-like Hamiltonian for the dynamics of atoms in an optical lattice and in presence of disorder and $H_\mathrm{C}$ the energy of the interaction with the cavity field. In detail, the optical lattice is composed of $K$ sites, we assume periodic boundary conditions and the atomic Hamiltonian reads

$$H_\mathrm{A} = -J \sum_j^K \left( b_{j+1}^\dagger b_j + \mathrm{H.c.} \right) + \sum_j^K E_j n_j + H_\mathrm{int}^\mathrm{F,B}. \tag{2}$$

where $b_j$ and $b_j^\dagger$ are the onsite annihilation and creation operators of a fermion or a boson at site $j = 1, \ldots, K$, (with $K+1$ identified with the first site), $J$ is the tunneling coefficient scaling

the nearest-neighbour hopping, $n_j = b_j^\dagger b_j$ is the occupation operator at site $j$, $E_j$ is the onsite energy at site $j$, and $H_{\text{int}}^{\text{F,B}}$ is the interaction term, which takes different forms depending on the quantum statistics of the atomic gas. For spinless fermions

$$H_{\text{int}}^{\text{F}} = U \sum_j^K n_j n_{j+1}, \tag{3}$$

and $U > 0$, whereas the first non–trivial interaction term in tight-binding expansion for bosons reads

$$H_{\text{int}}^{\text{B}} = U \sum_j^K n_j(n_j - 1). \tag{4}$$

In turn, the cavity-mediated long-range interactions take the form [28, 33, 37]:

$$H_{\text{C}} = -\frac{U_1}{K} \left( \sum_j^K (-1)^j n_j \right)^2 = -\frac{U_1}{K} \sum_{i,j}^K (-1)^{i+j} n_i n_j, \tag{5}$$

with $U_1 > 0$. This Hamiltonian term is here derived under the assumption that the wavelength of the cavity field equals the one of the electric field generating the optical lattice, see Appendix A. This interaction is proportional to the squared population imbalance, $H_{\text{C}} \propto -I(t)^2$, where

$$I(t) = \sum_i (-1)^i n_i \tag{6}$$

and the sign is due to the fact that the standing wave cavity mode with wave number $k$ takes value $\cos(kia) = (-1)^i$ at the optical lattice site centered at $x_i = ia$. Its expectation value is thus positive (negative) when the even (odd) sites of the cavity standing-wave mode are prevailingly occupied and it favours density-wave (DW) ordering [28, 33–35]. We determine the existence of the MBL phase by an exact diagonalization of the Hubbard Hamiltonian, Eq. (1), for spinless fermions. This situation is in fact more accessible to numerical analysis since the local Hilbert space has only dimension 2. We then only briefly show that analogous properties of the fermionic case are also found for bosons.

We finally note that the disorder in our model is in the onsite energy. Here we assume two cases. Throughout most of the work we make the theoretically elegant assumption that $E_j$ are uncorrelated random variables uniformly distributed in $[-W, W]$ interval, where $2W$ denotes the interval width. In Sec. 6 we then analyse the situation where $E_j$ is due to a quasi-periodic optical potential.

In the rest of this manuscript we report energies in units of $J$ and time in units of $1/J$.

## 3 Phase diagram

Energy level statistics encode an answer to the question of whether a disordered system is localized or ergodic and satisfies ETH. Level statistics of ergodic systems with (generalized) time reversal symmetry have properties akin to the Gaussian Orthogonal Ensembe (GOE) [43]. As the disorder strength increases and the system becomes localized, the level statistics becomes Poissonian [44, 45] (an accurate model for level statistics across the localization transition was recently proposed in Refs. [46, 47]). The level statistics can be characterized using the gap ratio. This is defined as

$$r_n = \min(\delta_n, \delta_{n+1})/\max(\delta_n, \delta_{n+1}), \tag{7}$$

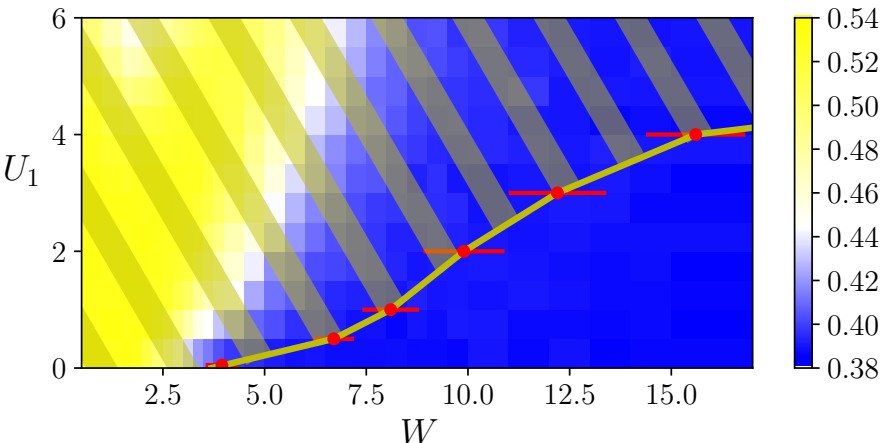

Figure 2: Contour plot of the mean gap ratio $\bar{r}$ as a function of random uniform disorder with amplitude $W$ and of the long-range interaction strength $U_1$. The mean gap ratio is determined in the center of the spectrum (for $\epsilon_n = (E_n - E_{\min})/(E_{\max} - E_{\min}) \approx 0.5$). The yellow and blue regions denote the ETH and the MBL phase, respectively, for $N = 8$ fermions, $K = 16$ lattice sites, and short-range interaction strength $U = 1$. A finite-size scaling analysis (see text and Fig. 3) places the boundary between ETH and MBL at the solid yellow line connecting the markers, giving the critical disorder amplitude $W_C(U_1)$. The error bars reported on the markers result from the comparison of finite-size scaling studies performed for $K = 16, 18, 20$ and $K = 14, 16, 18, 20$. The statistical errors for the color-map data points are well below 1% of their values.

with $\delta_n = E_n - E_{n-1}$ being the spacing between two consecutive eigenvalues [48]. Averaging over different energy levels within a certain interval as well as over disorder realizations results in the mean gap ratio, $\bar{r}$, that may be used to characterize the spectra. The mean gap ratio changes from $\bar{r} \approx 0.53$ in the ergodic regime [48, 49] to $\bar{r} \approx 0.39$ for a localized system and is thus a straightforward probe of the MBL transition especially as it does not require level unfolding, a tricky procedure for a many body system [50].

Figure 2 displays the contour plot of the mean gap ratio $\bar{r}$ in the $W - U_1$ phase diagram, namely, as a function of the disorder and of the long-range interaction strength. The colour code refers to the calculations performed for a gas of $N = 8$ fermions in a lattice with $K = 16$ sites, the gap ratio was first determined for 500 eigenvalues $E_n$ for which $\epsilon_n = (E_n - E_{\min})/(E_{\max} - E_{\min}) \approx 0.5$, where $E_{\max}, E_{\min}$ are respectively the largest and the smallest eigenvalue for given disorder realization, and then averaged over 400 disorder realizations. The statistics is sufficient to determine the mean gap ratio with an accuracy below 1% of its value. One clearly identifies two regions: (i) the yellow region, corresponding to $\bar{r} \approx 0.53$ where the system has GOE level statistics and is thus ergodic, and (ii) the blue region with $\bar{r} \approx 0.39$, where the system is MBL. The white stripe separates the ETH from the MBL regimes and gives the disorder strength at which $\bar{r} = 0.45$. This disorder strength depends on the system size and shifts to larger values as we increase the system size $K$.

Nevertheless, a finite-size scaling analysis suggests that the ergodic-MBL boundary converges to the yellow line connecting the red markers. The red markers are obtained as follows. We first consider the scaling form of the disorder strength given by

$$W \to (W - W_C(U_1))K^{1/\nu(U_1)}, \tag{8}$$

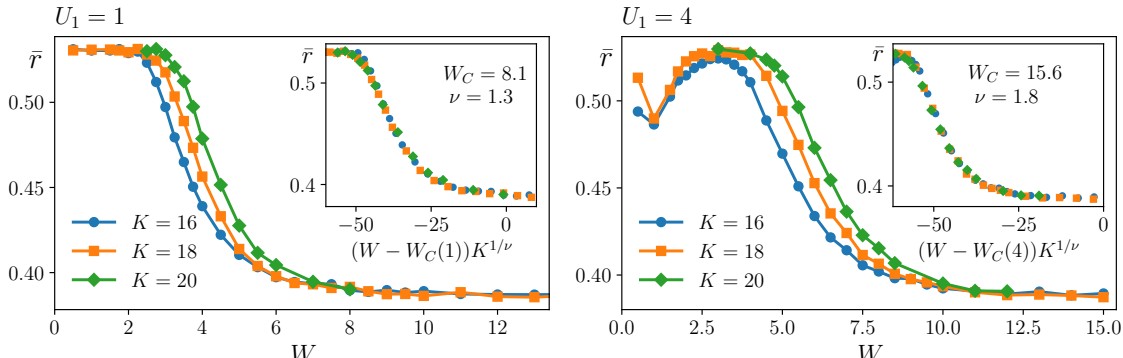

Figure 3: Onset: The mean gap ratio $\overline{r}$, in the center of the spectrum, is displayed as a function of the disorder amplitude $W$ and for $K = 16, 18, 20$ lattice sites. The left panel corresponds to $U_1 = 1$ , the right one to $U_1 = 4$. The insets display the data rescaled according to Eq. (8). Error bars are smaller than the marker's size. The universal functions $g_{U_1}[(W - W_C(U_1))K^{1/\nu}]$ are modeled by third order polynomials, points with $\overline{r} \in [0.392, 0.48]$ are taken into account in the finite-size scaling procedure.

where the critical disorder strength $W_C(U_1)$ and the exponent $\nu(U_1)$ depend on the long-range interaction strength $U_1$. We then consider the system sizes $K = 16, 18, 20$, which can be numerically simulated using the shift-invert technique, Ref. [51], implemented in Portable, Extensible Toolkit for Scientific Computation (PETSc) in Scalable Library for Eigenvalue Problem computations (SLEPc) setting, see Refs. [52, 53]. The onset of Fig. 3 displays the mean gap ratio $\bar{r}$ in the band center as a function of disorder strength for $U_1 = 1$ and $U_1 = 4$. The scaling (8) allows us to collapse the mean gap ratio for different system sizes $\overline{r}$ as a function of disorder strength $W$ onto universal curves $g_{U_1}[(W - W_C(U_1))K^{1/\nu}]$ with good accuracy. From these curves we extract the critical disorder strengths $W_C(U_1)$ for $U_1 \in \{0, 0.5, 1, 2, 3, 4\}$, which correspond to the markers in Fig. 2. From this ansatz we also determine the exponent $\nu(U_1)$. This increases with $U_1$ from the value $\nu(U_1 = 1) = 1.3(1)$ to $\nu(U_1 = 4) = 1.8(1)$. We remark that the scaling ansatz (8) is analogous to the one used for the standard MBL system with short-range interactions [54]. The fact that the same scaling seems to hold even in presence of long-range interactions suggests that the underlying physics of our system is similar.

So far the gap ratio analysis together with the finite-size scaling indicates the existence of a boundary between ergodic (ETH-like) and MBL phase. The mean gap ratio reaches the value characteristic for poissonian ensemble i.e. for integrable systems. This occurs for sufficiently large disorder values, strongly dependent on the long range interaction strength $U_1$ which realizes all-to-all couplings.

For completeness, let us note that the considered system possesses another non-ergodic phase at large values of the long range interaction strength $U_1 \gtrsim 10$ (not shown in Fig. 2). This regime emerges when the all-to-all coupling term dominates in the Hamiltonian leading to non-ergodic dynamics due to global interactions [55]. We discuss this regime in Appendix B considering here the regime of small $U_1$.

# 4 Dynamics of the system

Imagine we prepare the system in a well defined separable state $|\psi_0\rangle$. To probe the dynamics of the system we consider the time-dependent density correlation function

$$C(t) = D \sum_{i=1}^{K} (\bar{n}_i(t) - \bar{n})(\bar{n}_i(0) - \bar{n}), \tag{9}$$

where $\bar{n}$ is the average number of particles and $\bar{n}_i(t) = \langle\psi(t)|n_i|\psi(t)\rangle$ is evaluated over the evolved state $|\psi(t)\rangle = \exp(-iHt)|\psi_0\rangle$. Here, the constant $D$ warrants that $C(0) = 1$. According to ETH, an ergodic system loses the memory of the initial state and the correlation $C(t)$ decays to zero. Conversely, in the MBL phase the density correlation function $C(t)$ reaches a nonzero asymptotic value after a transient time of the order of few $J^{-1}$ (which is here set to unity) [54].

The second quantity with which we probe the dynamics is the bipartite entanglement entropy $S(t)$. This is obtained after splitting the lattice into two subsystems A and B and calculating the density matrix $\rho(t)$ of the subsystem $A$: $\rho(t) = \text{Tr}_B\{|\psi(t)\rangle\langle\psi(t)|\}$, where $\text{Tr}_B$ denotes the trace over subsystem B's degrees of freedom. The entanglement entropy is then defined as

$$S = -\sum_i \rho_{ii}(t) \log(\rho_{ii}(t)), \tag{10}$$

where $\rho_{ii}$ are Schmidt basis coefficients squared with $\sum_i \rho_{ii} = 1$ (see e.g. [56]). In systems with short-range interactions the logarithmic growth of the entanglement entropy $S(t)$ during the time evolution of the system is a hallmark of MBL [57,58] and can be understood within the picture of LIOMs [59,60]. Systems with strong long-range interactions, on the other hand, manifest dynamical properties typical of ergodicity breaking, such as the logarithmic growth of the entanglement entropy after quenches even in absence of disorder [55,61–63]. In order to single out the onset of localization and the effect of long-range interactions on the localization properties, in the following subsections we explore the transition between non-ergodic and ergodic regimes as a function of $U_1$ and for constant disorder amplitude $W = 8$. We then turn to the regime where the long-range interactions are a weak perturbation to the dynamics.

## 4.1 Density correlations and entanglement entropy

By inspecting the phase diagram (Fig. 2), one can see that for the disorder strength $W = 8$ the system is deep in the MBL phase for $U_1 = 0$. In fact, at $U_1 = 0$ Eq. (1) for fermions reduces to the XXZ Heisenberg spin chain and undergoes the ETH-MBL transition in the vicinity of $W_C(U_1 = 0) = 3.7$ [54]. This transition is accompanied by the appearance of non-vanishing values of the correlation function at the asymptotics, $C(t \to \infty) \neq 0$, as well as by the logarithmic growth of the entanglement entropy $S(t)$. We now consider a nonvanishing value of the long-range interaction strength, and in particular analyse the cases (i) $U_1 = 1$, (ii) $U_1 = 3$, and (iii) $U_1 = 5$. The latter two cases are both in the localized regime for $K = 16$, and yet they are delocalized in the thermodynamic limit according to the finite size scaling in Fig. 2. The case $U_1 = 1$, instead, corresponds to a MBL phase for all system sizes we consider as well as in the thermodynamic limit (as predicted by the finite-size scaling, see Fig. 2).

The time evolution of the correlation function $C(t)$ as well as of the entanglement entropy $S(t)$ is displayed in Fig. 4. We analyze system sizes $K = 16, 18, 20$ using the Chebyshev expansion technique for the time evolution (see Appendix C for details). For $U_1 = 1$ we observe the features characteristic of ergodicity breaking: the correlation function $C(t)$ acquires a stationary value which very weakly depends on the system size (compare with the left panel in Fig. 4). The entanglement entropy $S(t)$ (right panel) shows an increase with time which is

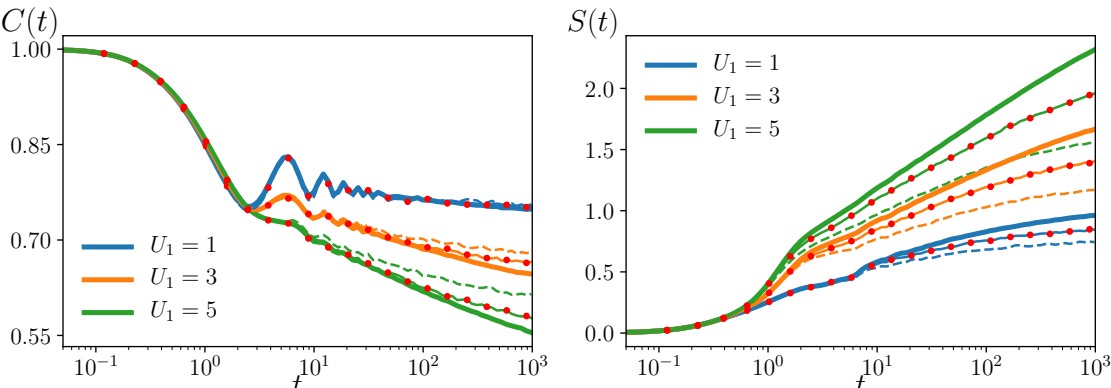

Figure 4: Ergodicity breaking for the system of spinless fermions (half-filling) with lattice sizes $K = 16, 18, 20$ (denoted respectively by dashed, dotted and thick lines), short-range interaction strength $U = 1$, disorder strength $W = 8$ and various long-range interaction strengths $U_1$. Left – the correlation function $C(t)$, Right – the entanglement entropy $S(t)$. The quantities are averaged over more than 2000 disorder realizations, starting from randomly chosen initial Fock states $|\psi_0\rangle$.

sublinear, and indeed seems weaker than logarithmic. As the strength of long-range interactions $U_1$ increases there appears a slow decay of the correlations $C(t)$ towards zero which becomes more pronounced as the system size $K$ is increased. This result suggest that the correlations $C(t)$ decay to zero in the thermodynamic limit, which is in agreement with the results of finite-size scaling. On the other hand the entanglement entropy $S(t)$ for $U_1 = 3$ and $U_1 = 5$ clearly grows logarithmically in time. Such a behavior is consistent with the picture of LIOMs and is believed to be a feature of MBL system. This seems to lead to an apparent paradox: In fact, while the dynamics of $C(t)$ suggests that large systems would be ergodic, at the same time, the entanglement entropy growth $S(t)$ shows no signs of delocalized behaviour. Yet the behaviour of $S(t)$ could also originate from the long-range nature of the interactions [55]. We observe, in particular, that the slope of $S(t)$ increases with $U_1$ and with the system size.

In order to gain insight we analyse in detail the behaviour of the entanglement entropy. Following Ref. [64] we express the entanglement entropy as the sum of two contributions $S(t) = S_P(t) + S_C(t)$, where $S_P(t)$ stems from particle number fluctuations between subsystems $A$ and $B$ and $S_C(t)$ is the entanglement entropy of different configurations of particles within the two subsystems. Let us denote by $p_n$ the probability of populating the $n$-particle sector in subsystem $A$ and by $\rho^{(n)}$ the corresponding block of the density matrix $\rho$ for subsystem $A$, such that $\rho = \sum_n p_n \rho^{(n)}$. A simple manipulation shows that Eq. (10) can be rewritten as

$$S(t) = -\sum_{n=0}^{N} p_n \log(p_n) - \sum_{n=0}^{N} p_n \sum_i \rho_{ii}^{(n)} \log(\rho_{ii}^{(n)}) \equiv S_P(t) + S_C(t). \tag{11}$$

The resulting behaviors of $S_P(t)$ and $S_C(t)$ are shown in Fig. 5. We first notice that exchange of particles between subsystems $A$ and $B$ occurs due to tunneling. As visible in left panel of Fig. 5 $S_P(t)$ grows significantly at a time scale of few tunneling times, independently of the value of $U_1$ and of the system size. After this transient, its behaviour depends significantly on $U_1$ and on the system size. In particular, for $U_1 = 1$ it grows very slowly with time and weakly depends on the system size, hinting towards a strong suppression of particle number fluctuations. For $U_1 = 5$, instead, it has a clear logarithmic growth in time and a significant dependence on the system size. The former case is a standard MBL behavior [64]: the logarithmic growth

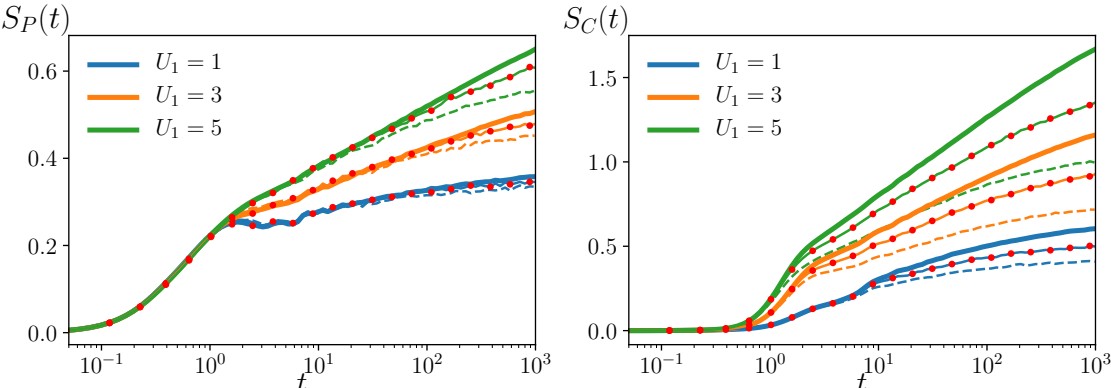

Figure 5: Particle (left) and configuration (right) entanglement entropy ($S_P(t)$ and $S_C(t)$ respectively) for the system of spinless fermions (half-filling) with lattice sizes $K = 16, 18, 20$ (denoted respectively by dashed, dotted and thick lines), short-range interaction strength $U = 1$, disorder strength $W = 8$ and long-range interaction strengths $U_1 = 1, 3, 5$.

of $S(t)$ is mainly due to the increase in the configuration entropy $S_C(t)$. The latter behavior, in which $S_P(t)$ grows logarithmically in time enabling also faster and faster growth of $S_C(t)$ leads eventually to thermalization. The dynamics at large $U_1$ thus points towards ergodicity for larger sizes, in agreement with the finite-size scaling analysis. Yet, it is so slow that the decay time of the correlation function $C(t)$ is much slower than in the ergodic regime at small $W$ and $U_1 = 0$. We remark that in standard models with short-range interactions deeply in the MBL phase, the time evolution can be efficiently simulated to large times ($\approx 10^3$) and for large systems sizes ($K \approx 10^3$) [65] with time-dependent density matrix renormalization group related approaches [66]. Such approaches are ruled out by the infinite interaction range of our model. We note that algorithms based on time dependent variation principle [67,68] could in principle tackle large system sizes. We leave this task for future work and here consider small sizes, amenable to diagonalization-like treatments.

## 4.2 Weak long-range interactions

We now analyse the role of the long-range interactions on MBL by considering the limit in which $U_1$ is sufficiently weak with respect to the tunneling rate. We first focus on the case $U = 0$, when the particles solely interact via the long-range interactions. For $U_1 \ll 1$ we expect that the dynamics is first dominated by the hopping, and only on a much longer time scale it is going to be visibly affected by $U_1$. The left panel of Fig. 6 displays the entanglement entropy $S(t)$ for different values of $U_1$, ranging from $1.6 \times 10^{-3}$ till 0.8. The entanglement entropy first rapidly grows during an initial transient, which is the same for all considered values of $U_1$ and is of the order of $1/J$ (which is here the unit of time). After this transient $S(t)$ saturates to a value for a time interval, up to a time scale $T_1 \approx 1/U_1$. We understand this behavior as the system being in the Anderson localization regime, since for this time scale the dynamics is the one of non-interacting particles. After $T_1$ the long-range interactions start to significantly affect the dynamics and the entanglement entropy $S(t)$ increases approximately linearly with time, till it saturates. The corresponding saturation value depends on the given system size, on the strength of disorder $W$, and on the long-range interaction strength $U_1$, being, however, much lower, than the maximal (ergodic) value for a given system size.

The behaviour of the entanglement entropy $S(t)$ for $U = 1$ (thus when $U = J$) is shown

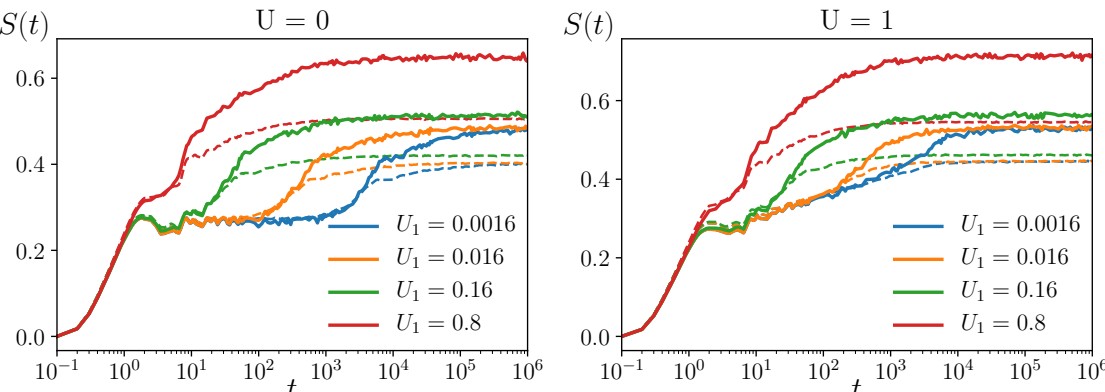

Figure 6: Entanglement entropy as a function of time for $U = 0$ (left) and $U = 1$ (right), $W = 8$ and for different values of $U_1$ (see color code in the legenda). The dashed and thick lines correspond to $K = 12$ and $K = 16$, respectively.

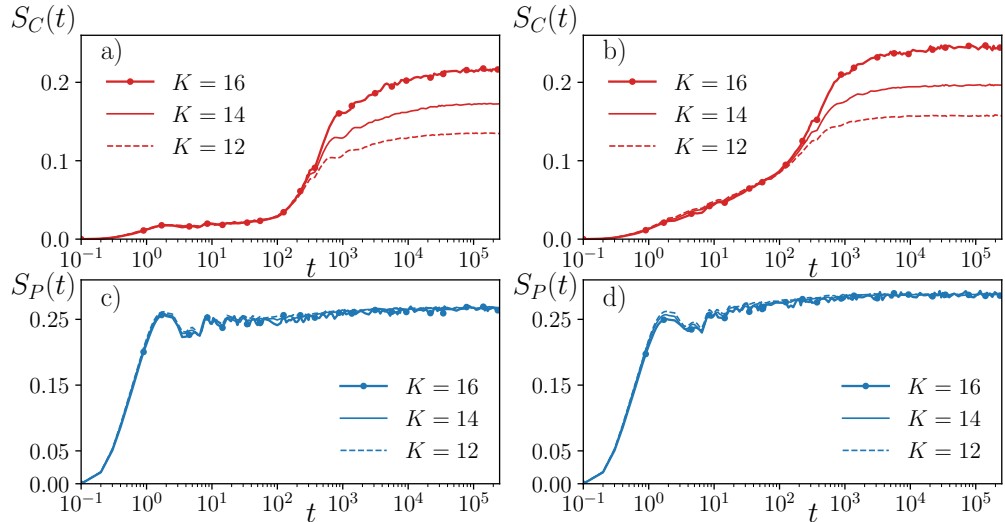

Figure 7: Configuration entanglement entropy $S_C(t)$ [top row, panels a) and b)] and the particle entropy $S_P(t)$ [bottom row, panels c) and d)] as a function of time in the localized regime ($W = 10$) of the model with $U = 0$ and $U_1 = 0.016$ corresponding to the Anderson case perturbed by the long-range interactions [left column, panels a) and c)]. The plots in the right column [panels b) and d)] with $U = 1$ correspond to long-range interaction perturbation of the MBL case. The effect of small $U_1$ in both cases is quite similar. Thick, dashed and thin lines correspond to $K = 16, 14$ and $K = 12$ respectively.

in the right panel of Fig. 6. The time scale separation allows us to identify two behaviours characterizing the entanglement growth: the growth which goes logarithmic in time, as for a standard MBL system, and the rapid transient of $S(t)$ at the time scale $T_1$ due to the all-to-all coupling. The saturation values of the entanglement entropy seems to be only weakly dependent on the interaction strength $U$. This leads us to conjecture that the origin of the nonergodic dynamics here is also MBL and it arises from a quasi-degenerate manifold of states coupled by the long-range interactions.

In order to test this conjecture, in Fig. 7 we separately plot the particle and the configuration entanglement entropy as a function of time and corresponding to the curves of Fig. 6 with $U_1 = 0.016$ and $U = 0$ and $U = 1$. For both values of $U$ the particle entanglement entropy $S_P(t)$ increases rapidly as the initial occupation of the lattice sites spreads due to tunneling. This transient dynamics occurs on a time scale of the order of $1/J$, after which the particle number fluctuations change only marginally: observe that the associated $S_P(t)$ is approximately independent of the considered system sizes $K$. Instead, the configuration entanglement entropy $S_C(t)$ shows different behaviors as a function of $U$ and of $K$. The dynamics of $S_C(t)$ can be characterized by the time scale $T_1 \sim 1/U_1$: for $t \lesssim T_1$ and for $U = 0$ it is roughly constant, while for $U = 1$ it grows logarithmically with time. After $T_1$ the configuration entanglement entropy rapidly grows and then saturates to a value which increases with the system size. We understand this increase as the number of accessible configurations grows with $K$. On the basis of this analysis we conclude that the dynamics for $U = 0$ and small $U_1$ is a textbook case of Anderson localization perturbed by weak long-range interactions. The long-range interactions couple only states that are closely spaced in energy and that are localized in different regions of space. Thus long-range interactions lead to a spread of the initial state among relatively few localized eigenstates, the corresponding dynamics is strongly nonergodic. The analogies shared between the dynamics at $U = 0$ and $U = 1$ for small $U_1$ suggest that for $U \neq 0$ the dynamics is MBL-like. MBL is perturbed by the long-range interactions, which strongly couple the quasi-degenerate manifold of localized states and at the same time the dynamics remains nonergodic.

We explore the properties of this peculiar MBL phase by using a LIOM picture. We first recall that the Hamiltonian of a generic (fully) many-body localized system can be expressed as [59, 60]

$$H = \sum_{i=1}^{K} J_i^{(1)} \tau_i^z + \sum_{i,j=1}^{K} J_{ij}^{(2)} \tau_i^z \tau_j^z + \sum_{i,j,k=1}^{K} J_{ijk}^{(3)} \tau_i^z \tau_j^z \tau_k^z + ..., \tag{12}$$

where $\tau_i^z$ are quasi-local operators knows as LIOMs or l-bits. They can be thought of as dressed occupation number operators as $\tau_i^z = U^\dagger n_i U$ where $U$ is a quasi local unitary transformation. The couplings $J_{ij}$ fall off exponentially with the distance between interacting l-bits as

$$J_{ij}^{(2)} = J_0 e^{-|i-j|/\xi}, \tag{13}$$

where $\xi$ is the localization length (a similar relation holds for higher order couplings $J_{ijk}^{(3)}, ...$). It has been analytically shown [69] that the l-bit model (12) leads to a logarithmic growth of the Renyi-2 entropy $S_2(t) = -\log \text{Tr}\rho^2$ (which for large times and large system sizes behaves essentially as the von-Neumann entanglement entropy $S(t)$) assuming that the initial state is an equal superposition of all Fock states. Moreover, to observe the logarithmic growth of the Renyi entropy $S_2(t)$ it suffices to keep only the coupling coefficients $J_{ij}^{(2)}$, Eq. (13), setting the higher-order couplings $J_{ijk}^{(3)}, ...$ equal to zero.

In Fig. 8 we show that the Renyi entropy of the l-bit Hamiltonian (12) reproduces the behaviour of the configuration entropy $S_C(t)$ for the extended Hubbard model with spinless fermions. Specifically, assuming the exponential decay of the coupling terms $J_{ij}^{(2)}$, Eq. (13), we reproduce the logarithmic growth of entanglement entropy characteristic of standard MBL. In order to reproduce the rapid growth of $S_C(t)$, we introduce long-range couplings between LIOMs according to

$$\tilde{J}_{ij}^{(2)} = J_0 e^{-|i-j|/\xi} + \frac{J_1}{K}(-1)^{i+j} r_{ij}, \tag{14}$$

where the term $J_1/K(-1)^{i+j} r_{ij}$ mimics the coupling experienced by l-bits caused by the long-range interaction term $H_C$ (5) and $r_{ij} \in [0, 1]$ is a random variable which models the overlap

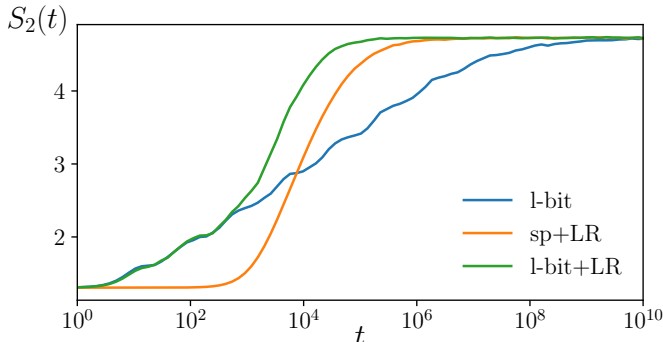

Figure 8: Renyi entropy $S_2(t)$ as a function of time for the $l-bit$ model of Eq. (12) with (i) $J_0 > 0$, $J_1 = 0$ (l-bit), (ii) $J_0 > 0$, $J_1 > 0$ (l-bit+$LR$), (iii) $J_0 = 0$, $J_1 > 0$ (sp+$LR$), see legenda. The system size is $K = 16$ and the initial state (following [69]) is $|\psi\rangle = \sum_{i=1}^{N_H} |FS_i\rangle / \sqrt{N_H}$ where $|FS_i\rangle$ is $i$-th state of the Fock basis and $N_H$ is dimension of the Hilbert space. The value of Renyi entropy $S_2(t)$ corresponds in this model to $S_C(t)$ of the extended Hubbard model with spinless fermions.

of $\tau_i^z$ and $n_i$. If we set $J_0 = J_1 = 0$ in Eq. (14), then there is no growth of entanglement entropy in the l-bit model which corresponds to no growth of configuration entropy – a situation characteristic of Anderson localization. Introducing non-zero $J_0$ in the model (12) we obtain the logarithmic growth of $S_2(t)$ – the hallmark of MBL. Setting then $J_1$ to a finite, non-vanishing value one gets a rapid growth of entanglement entropy which starts at a certain time scale set by $J_1$: After this growth $S_2(t)$ saturates at the same value as in the case $J_1 = 0$. All of those feature are in qualitative agreement with the growth of configuration entropy $S_C(t)$ for our system at $U_1 = 0.016$ and $U = 1$.

The analysis of this Section strictly applies for very small values of $U_1$, such that the time scale $T_1 = 1/U_1$ is much larger than the time scale $J^{-1}$ set by the tunneling rate. As visible in Fig. 6, this separation of time scales takes place as long as $U_1 \lesssim 0.16$. The saturation value of the entanglement entropy $S(t)$ is already slightly larger for $U_1 = 0.16$ than for smaller values of $U_1$ meaning that a slight modification of structure of the LIOMs (possibly an increase of the support of $\tau_i^z$) happened. For larger $U_1 = 0.8$, the saturation value of $S(t)$ is significantly larger. On the basis of the discussion so far, in particular of the studies in Sec. 3, we conclude that the system is still MBL, however, the properties of l-bits $\tau_i^z$ are significantly affected by the long-range interactions.

In summary, we have shown that the cavity mediated long-range interactions lead to non-ergodic behavior of the system, which in presence of strong disorder may be interpreted as MBL. However, the logarithmic growth of entanglement entropy, a hallmark of standard MBL is missing in our model, at least for very small $U_1$. The observed ergodicity breaking exhibits novel features, such as the rapid growth of entanglement entropy which we attribute to the long-range interactions. When the effect of the long-range interactions can be separated from the short-range interactions in the dynamics, then the entanglement entropy increases in good approximation linearly in time. Interestingly the entanglement entropy is still bounded by a constant which only moderately changes with $U_1$ as long as the system is in the localized phase. On the other hand, the dynamics and in particular the growth of the configuration entropy may be explained using the language of LIOMs as described above, the system is integrable (as the gap ratio analysis reveals poissonian statistics) and the entropy growth saturates to smaller values than those found for ergodic dynamics. For these reasons we consider the behavior observed as a nonstandard variant of MBL. Let us also note that the observed phenomenon is

different from the non-ergodic phase observed for systems with single particle mobility edge [70–73] where long-range interactions are not involved.

# 5  Detection of ergodicity breaking: Light at the cavity-output

The dynamics of the system can be experimentally revealed via a site-resolved measurement of $n_i$, which can be performed in cold atoms experiments [7] and which allows one to reconstruct the correlation function $C(t)$. In this section we argue that the cavity setup can allow one to measure the breakdown of ergodicity by photo-detection of the light at the cavity output. The emitted light, in fact, is scattered by the atoms and thus contains the information on their density distribution. In particular, the electric field amplitude, which we denote by $E_{out}$, is directly proportional to the expectation value of population imbalance $I(t) = \sum_i (-1)^i n_i$, Eq. (6) in the dispersive cavity regime (see Appendix A and [31,32,37,74,75]):

$$E_{out}(t) \propto \langle I(t) \rangle, \qquad (15)$$

which can be measured via heterodyne detection [74] (Recall that an observable similar to $I(t)$ was employed in Ref. [7] to demonstrate MBL for a system of fermions in the optical disordered lattice). This measurement introduces projection noise that affects the atomic dynamics. Nevertheless, as we detail in Appendix A, this noise is negligible in the limit we consider, where the cavity field dynamics occurs on a much faster time scale than the atomic motion. The information extracted from the light at the cavity output has been used in this fashion to extract information about ground (or metastable) state phases of similar systems [31,37,75].

Below we discuss the time evolution of the imbalance $\langle I \rangle$ and of its square $\langle I^2 \rangle$ (corresponding to the light intensity) for two states in which the system can be initially prepared. We first consider a density-wave like state $|DW_{10}\rangle = |101010..\rangle$, with odd sites occupied and even sites empty. This state maximizes $\langle I \rangle$ for the fractional density $\bar{n} = 1/2$ and minimizes the energies of both short-range and long-range interactions. In the absence of disorder $|DW_{10}\rangle$ is an eigenstate of the Hamiltonian in the atomic limit [34]. Starting from this state, in the following we investigate to what extent the experimentally accessible time evolution of the population imbalance $\langle I \rangle$ allows one to probe ergodicity and its breaking in the system. In this analysis one shall keep in mind that for $U_1 \gtrsim W$ the state $|DW_{10}\rangle$ has a significant overlap with the ground state, and its energy thus gets closer to the ground state energy as the ratio $U_1/W$ is increased. Figure 9 diplays the time evolution of $\langle I \rangle$ for the system initially prepared in $|DW_{10}\rangle$ state. The dynamics is shown for $W = 0.5$ (upper panels) and $W = 10$ (lower panels). These two disorder strengths $W$ correspond to the ergodic and the MBL regimes, respectively, for the considered values of $U_1$. For comparison, we also display the dynamics of the correlation function $C$ for the same parameters, but when the initial state is a random Fock state with the same density. In the ergodic regime ($W = 0.5$), the correlation function $C$ decays to zero signifying total relaxation of the initial density profile regardless of the value of $U_1$. The dynamics of the population imbalance for the initial state $|DW_{10}\rangle$ depends strongly on the ratio $U_1/W$. For $W = 0.5$ and $U_1 = 0.16$, the imbalance $\langle I \rangle$ decays to zero and is a valid probe of the ergodic properties of the system. For the same disorder amplitude and larger values of $U_1$, instead, $\langle I \rangle$ saturates at a constant value, from which we infer that the state $|DW_{10}\rangle$ has already significant overlap with the ground state of the system. In this regime, the time evolution of $\langle I \rangle$ is not a valid probe of ergodicity of a typical state of the system.

For strong disorder, $W = 10$, on the other hand, the information about localization properties of the system provided by the correlation function $C$ is also visible in the time evolution of the imbalance $\langle I(t) \rangle$, even for $U_1 = 3.2$. This behaviour is due to the fact that the energy spectrum of the system is much broader, such that even for values of the interaction strength

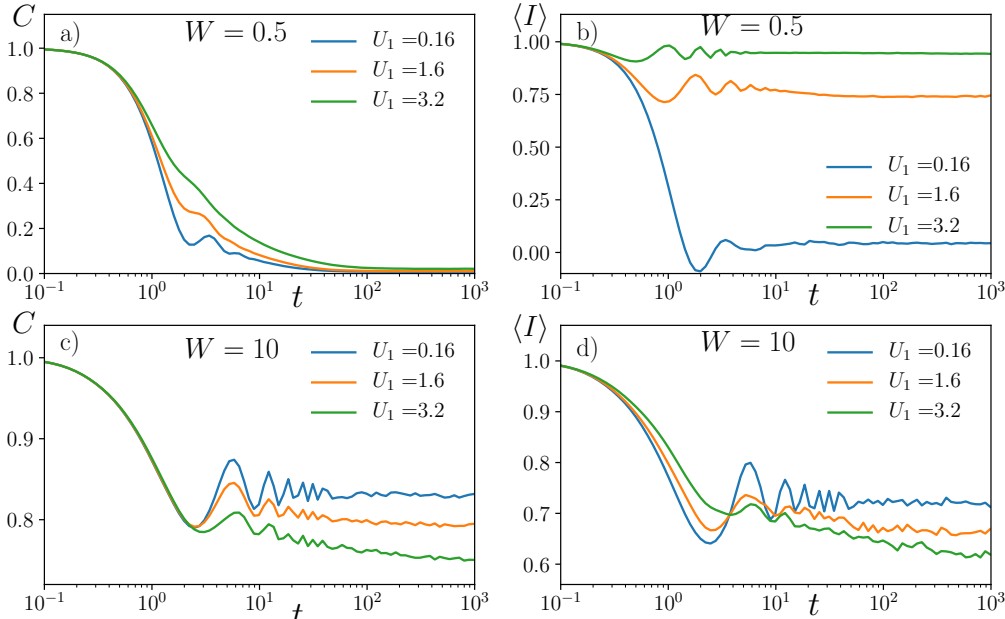

Figure 9: Time evolution of the correlation function $C(t)$, Eq. (9), [panels a) and c)] and the imbalance $\langle I \rangle$, Eq. (6) [ panels b) and d)] for different values of long-range interactions $U_1$. Top row [panels a) and b)] corresponds to small disorder $W = 0.5$ for which MBL is not expected. Yet, for $U_1 = 1.6, 3.2$ the imbalance saturates to finite values. The initial state of $C(t)$ is a random Fock state, for $\langle I \rangle$ is $|DW_{10}\rangle$. For strong disorder [bottom row, panels c) and d) ], where MBL is expected, the finite values of the population imbalance correspond to the saturation of $C(t)$. The data have been calculated $K = 16$ and averaged over 1000 disorder realizations (starting time evolution from different randomly chosen Fock state in calculation of $C(t)$).

as large as $U_1 = 3.2$ the state $|DW_{10}\rangle$ is still in a region of the spectrum with a relatively large density of states.

We now consider the initial state $|DW_2\rangle = |11001100...\rangle$, where the initial site occupations form a pattern of singly-occupied and empty doublets of lattice sites. State $|DW_2\rangle$ is an eigenstate of the imbalance operator $I$ at the eigenvalue 0, therefore its energy lies about the band center. Figure 10 displays the time evolution of squared population imbalance $\langle I^2 \rangle$ when the initial state is $|DW_2\rangle = |11001100...\rangle$ and for different values of the disorder amplitude: $W = 1, 3, 7, 10$. The upper panels report, for comparison, the correlation function $C$ for the corresponding values of the disorder amplitude but when the initial state is a random Fock state. The dynamics of the density-density correlation function $C$ shows that the system is ergodic for $W = 1, 3$ with correlations eventually decaying to zero. For $W = 7, 10$, instead, $C$ saturates to non-zero values at large times signalling MBL. The behaviours for vanishing long-range interactions and for $U_1 = 2$ are similar, see Fig. 10 a) and b). If the system is ergodic, the "intensity" $\langle I^2 \rangle$ increases from zero over a time scale of few $J^{-1}$ till it reaches a nonvanishing saturation value which depends on the disorder strength $W$. As visible in Fig. 10 c), the saturation value is maximal in the ergodic phase at $W = 1$ and is smaller in the MBL phase at $W = 7, 10$. The saturation value of $\langle I^2 \rangle$ thus reflects the ergodicity of the system dynamics. The saturation values of $\langle I^2 \rangle$ still point towards non-ergodicity for $U_1 = 2$ at $W = 7, 10$, consistently with the value of mean gap ratio $\bar{r}$ for system size $K = 16$, see Fig. 2. Note that the saturation values of $\langle I^2 \rangle$ at $U_1 = 2$ are larger than the ones reached for $U_1 = 0$ at the same

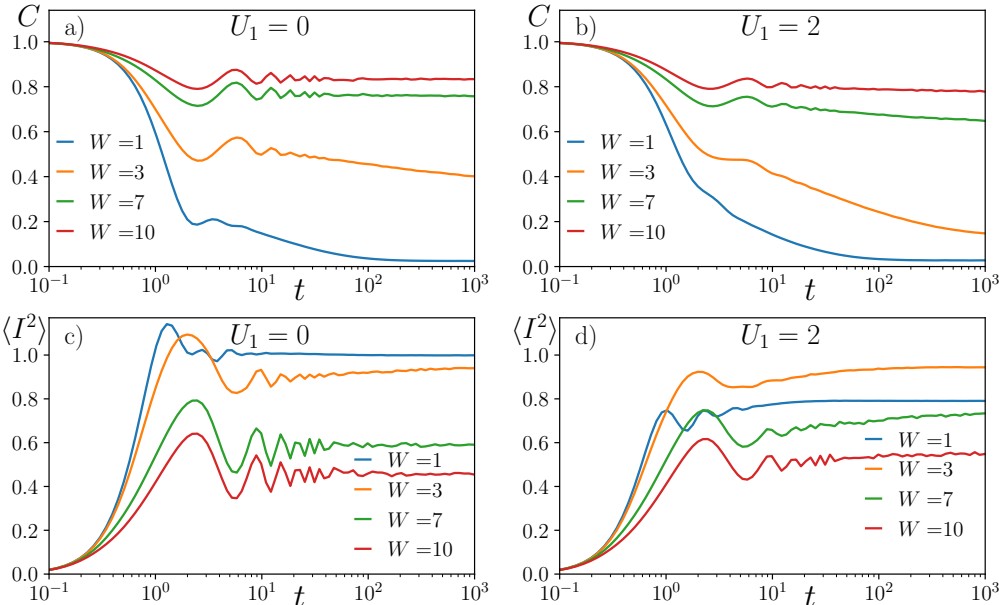

Figure 10: Time evolution of the correlation functions (top panels) and of $\langle I^2(t) \rangle$, corresponding to the intensity of the emitted light, (lower panels) for $U_1 = 0$ (a),(c) and $U_1 = 2$ (b),(d), and for different values of $W$. The initial state of the upper panels is a random Fock state, for the lower panels state $|DW_2\rangle$. With increasing disorder strength the correlations saturate to larger values in the MBL regime - that is associated with a decrease of $\langle I^2 \rangle$. The normalization of $\langle I^2 \rangle$ is chosen so that it saturates to unity at large times for $W = 1$ and $U_1 = 0$. The data have been calculated $K = 16$ and averaged over 1000 disorder realizations (starting time evolution from different randomly chosen Fock state in calculation of $C(t)$).

disorder strength. This behaviour is consistent with the intuition that long-range interactions tend to delocalize. In Fig. 10(d) one observes that the saturation value of $\langle I^2 \rangle$ is lower for $W = 1$ than for $W = 3$. This suggests that the subspace of the Hilbert space accessible during the time evolution from $|DW_2\rangle$ state is constrained by long-range interactions at small disorder strength $W = 1$, and that this constraint becomes weaker at $W = 3$.

In summary, we have argued that the measurement of light emitted by the cavity can allow one to determine the localization properties of the system by starting from well defined states. This probe of of ergodicity breaking is an appealing alternative to the band mapping technique of Refs. [76, 77] used in standard population imbalance measurements.

## 6 Quasi-random disorder

Up till now we have considered a random on-site disorder. Such a situation may be realized by an off-resonant pumping laser with a random intensity distribution. This laser shall drive an atomic transition that does not scatter into the cavity field, and thus generate a random a.c. Stark shift of the onsite energy. The disorder may. also be realized with the setups of Ref. [37, 78] by introducing additional weak laser beams creating optical lattices, whose periodicity is incommensurate with the cavity lattice periodicity. This setup creates quasi-random disorder analogously to the experiment of Ref. [7]. We now analyse localization in such a case i.e. when the onsite energy is due to the contribution of an incommensurate

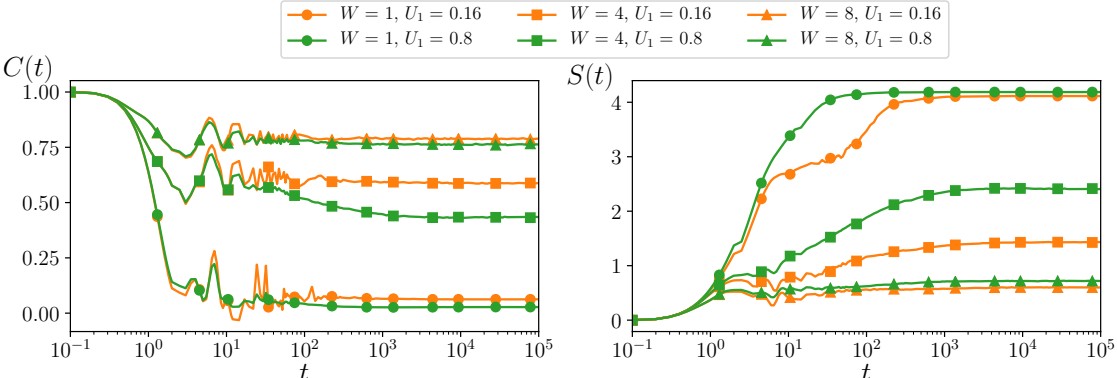

Figure 11: Time evolution of the density correlation function $C(t)$ and of the entanglement entropy $S(t)$ for $N = 7$ spinless fermions on $K = 14$ sites for the extended Hubbard model with quasirandom disorder, Eq. (16), for $U = 0$ and the long-range interactions strengths $U_1 = 0.16, 0.8$. The results are averaged over 1000 values of the phase $\phi$ in the interval $[0, 2\pi]$, the initial state is a Fock state with random site occupation and $N = 7$.

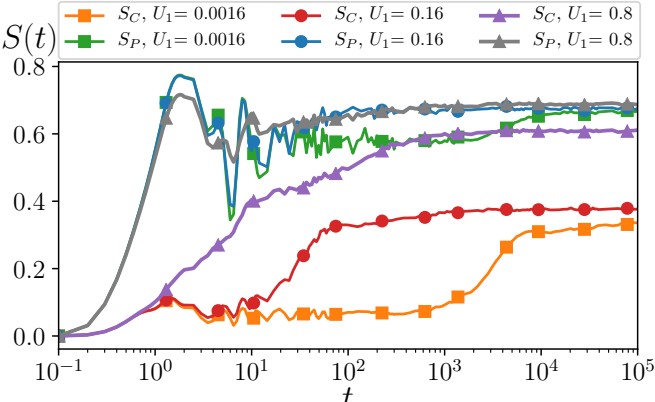

Figure 12: Time evolution of the configurational entanglement entropy $S_C(t)$ and of the particle entanglement entropy $S_P(t)$ for $N = 7$ spinless fermions on $K = 14$ sites with quasirandom disorder. The parameters are $U = 0$ and $W = 8$. The results are averaged over the phase $\phi$ in the interval $[0, 2\pi]$, the initial state is a Fock state with random site occupation and $N = 7$.

periodic potential, namely,

$$E_j = W \cos(2\pi\beta j + \phi). \tag{16}$$

Here, $W$ plays a role of disorder amplitude, $\beta$ is the ratio of the two lattice periodicities (we take $\beta = (\sqrt{(5)} - 1)/2)$ and the value of the phase $\phi \in [0, 2\pi]$ determines the disorder realization. Quasiperiodic potentials such as the one of Eq. (16) have been employed in a number of experiments investigating MBL in optical lattices [7, 73, 79, 80]. It is well known that the properties of the MBL transition for purely random disorder differ substantially from the ones of quasi-periodic disorder, as it was shown by the analysis of the entanglement entropy [81] and of the gap ratio [47]. Yet, while the important aspects of the transition itself are different, the ergodic and MBL phases are similar in the two settings, which is the reason why the seminal observation of MBL [7] was feasible in a setup with quasiperiodic disorder. In order to show that similar features are found in our extended Hubbard model, we here discuss

the dynamics of spinless fermions when the onsite energy in Eq. (2) is given by Eq. (16).

The time evolution of the correlation function $C(t)$ and of the entanglement entropy $S(t)$ are displayed in Fig. 11. Two distinct regimes can be here identified: (i) the ergodic phase at $W = 1$ in which the correlations $C(t)$ rapidly decays to zero and entanglement spreads ballistically and (ii) the MBL phase at $W = 8$ characterized by an asymptotic non-vanishing value of the correlation function $C(t)$ as well as by the logarithmic growth of entanglement entropy. The intermediate disorder strength $W = 4$ corresponds to the regime in which the localized-to-ergodic transition takes place as the strength of all-to-all coupling $U_1$ is increased. To provide further evidence that the physics is similar for both random and quasiperiodic disorders, we calculate the time evolution of the configurational entanglement entropy $S_C(t)$ and of the particle entanglement entropy $S_P(t)$. The results are presented in Fig. 12. Similarly to the case of random disorder, small values of $U_1$ lead to rapid growth of $S_C(t)$ at the time scale $T_1 = 1/U_1$. For $U_1 = 0.16$, $S_C(t)$ reaches a larger asymptotic value. Further increase of the long-range interaction strength $U_1$ leads to the logarithmic growth of $S_C(t)$ with time, the entanglement entropy saturates at much larger value.

We thus predict that the properties of the MBL phase of spinless fermions with quasiperiodic disorder and in the presence of cavity-mediated all-to-all interactions are analogous to the ones found when the disorder is instead random.

## 7  Bosons in the cavity

Many-body localization in bosonic systems was studied numerically in [9, 10, 82] as well as in experimental realizations [64, 83]. The essential features of the MBL phase in bosonic models are analogous to the system of spinless fermions (or spins). However, the possibility that the lattice site occupations exceeds unity leads to the natural appearance of many-body mobility edges at higher energy densities. Motivated by the fact that several experiments with cold atoms in optical resonators [37, 78] are performed with bosons, here we briefly discuss signatures of the MBL in the extended Bose-Hubbard Hamiltonian:

$$H = -J \sum_j^K \left( b_{j+1}^\dagger b_j + \text{H.c.} \right) + \sum_j^K E_j n_j + U \sum_j^K n_j(n_j - 1) - \frac{U_1}{K} \sum_{i,j}^K (-1)^{i+j} n_i n_j. \qquad (17)$$

Figure 13 displays the phase diagram for a lattice consisting of $K = 8$ sites at unit filling. The diagram reports the mean gap ratio $\bar{r}$, obtained from the gap ratio $r$ averaged over the states at the center of the spectrum $\epsilon \approx 0.5$. It shows that for a given value of the all-to-all coupling $U_1$ there exists a disorder strength sufficient to localize the system. The result is similar to the phase diagram of spinless fermions in Fig. 2. We note however, that the values of disorder required to induce many-body localization in the bosonic system of Eq. (17) are larger than the ones of the fermionic counterpart.

To provide further insight into the physics of the MBL phase in all-connected bosonic system we calculate the bipartite entanglement entropy during the course of time evolution of the system – c.f. Fig. 14. The features are overall similar to the ones found for spinless fermions and thus seem to not be limited to small Hilbert spaces per site as for fermions.

## 8  Conclusions

In this work we have analysed the occurrence of many-body localization in a system of particles with all-to-all interactions. The dynamics is described by an extended Hubbard model, where the onsite energy follows a random distribution. Our study is numerical and is based on exact diagonalization as well as on methods for sparse Hamiltonian matrices. By means of

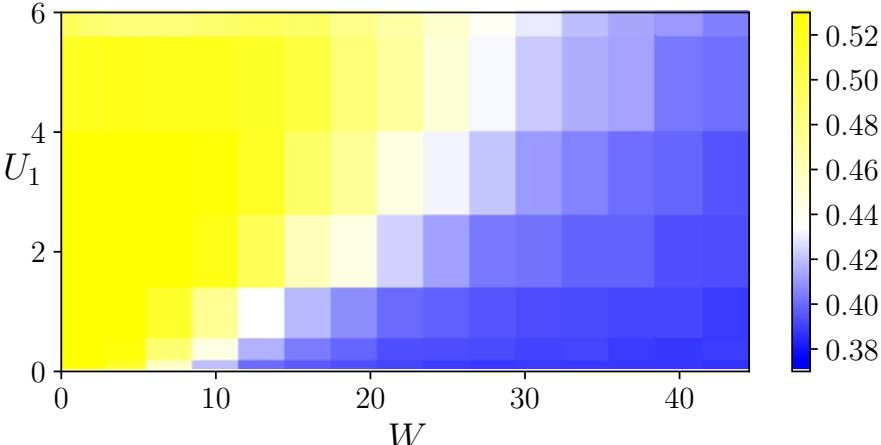

Figure 13: Contour plot of the average gap ratio $\overline{r}$ as a function of $W$ and $U_1$. The average gap ratio is calculated at the center of spectrum ($\epsilon = 0.5$) for $N = 8$ bosons on $K = 8$ lattice sites. The on-site interaction strength is $U = 1$, the results are averaged over 160 disorder realizations.

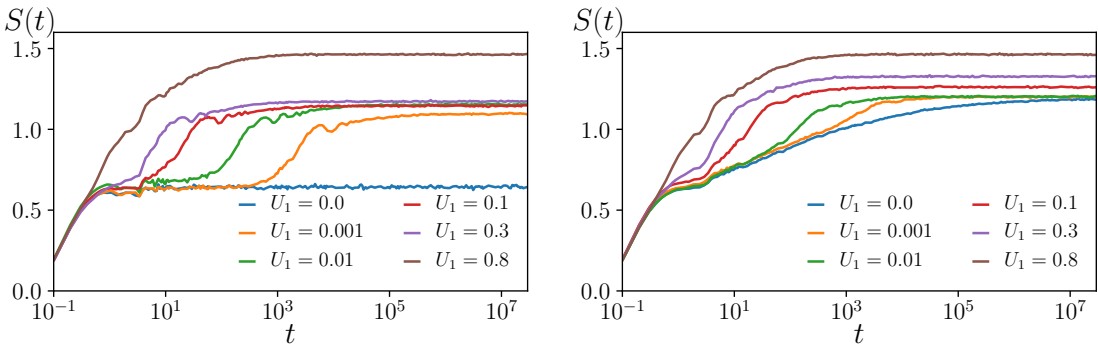

Figure 14: Bipartite entanglement entropy $S(t)$ as a function of time for $N = 8$ bosons on $K = 8$ lattice sites. The left panel corresponds to $U = 0$, the right panel to $U = 1$, the initial state is a random Fock state. The results are averaged over 2000 disorder realizations.

finite-size scaling we have shown that the MBL phase is indeed present in the system and that the transition to the ergodic phase occurs at disorder amplitudes whose critical value increases with the long-range interaction strength. The all-to-all interactions affect the spreading of the entanglement during the time evolution: the growth of entanglement entropy is faster than logarithmic in the MBL phase. It is instead linear when the tunneling coefficient is much larger than long-range interaction strength. Nevertheless, the saturation value of entanglement entropy is constrained by particle number fluctuations between partitions of the system. A fixed distribution of particle number between subsystems is attained during the course of time evolution even in presence of all-to-all interactions and hence, the dynamics is non-ergodic. We have shown that the features of the MBL phase can be qualitatively understood within the picture of LIOMs with long-range couplings. The interplay between disorder and long-range interactions results in transition from the MBL to an ergodic phase. The characteristic features of the observed ETH-MBL transition seem to be independent of the quantum statistics, as the

numerical analysis for finite systems of spinless fermions and of bosons show. Similar features are observed both for true random disorder as well as for quasi-periodic potential.

This dynamics can be observed experimentally with quantum gases in an optical resonator. For these systems, indeed, photo-detection of the light at the cavity output can allow one to probe the gas phase and thus provide evidence of ergodicity breaking.

We finally remark that MBL phase in presence of global coupling to $d$ dimensional system was found in [24], particular scaling of $d$ with system size leads in high frequency limit to Hamiltonian similar to the one considered in this work.

## Acknowledgements

This work was performed with the support of EU via Horizon2020 FET project QUIC (nr. 641122). Numerical results were obtained with the help of PL-Grid Infrastructure. We acknowledge support of the National Science Centre (PL) via project No.2015/19/B/ST2/01028 (K.B. and P.S.), 2018/28/T/ST2/00401 (Etiuda scholarship – P.S.) and the QuantERA QTFLAG programme No. 2017/25/Z/ST2/03029 (J.Z.). G.M. acknowledges support by the German Research Foundation (DFG, Priority Program No. 1929 GiRyd) and by the German Ministry of Education and Research (BMBF) via the QuantERA project NAQUAS is acknowledged. Projects NAQUAS and QTFLAG have received funding from the QuantERA ERA-NET Cofund in Quantum Technologies implemented within the European Union's Horizon 2020 Programme.

## A  Derivation of the Hubbard Hamiltonian

Here we present the summary of the derivation of Hamiltonian ((1)-(5)). The details in different variants of the model are discussed in e.g. Refs. [26, 28, 31, 33–35].

We consider $N$ atoms of mass $m$, they are confined within an optical cavity in a quasi-one-dimensional configuration colinear with a one-dimensional optical lattice. The lattice oriented along the cavity axis has the same wave number $k = 2\pi/\lambda$ as the cavity field. The atoms are prepared in the electronic ground state $|1\rangle$, the electric dipole transition $|1\rangle \to |2\rangle$ couples dispersively to the cavity and to the transverse laser with Rabi frequency $\Omega_z(x)$, see Fig. 1. Atoms and cavity field are treated in second quantization: $\hat{a}$ and $\hat{a}^\dagger$ are the annihilation and creation operators of a cavity photon, respectively, and obey the commutation relation $[\hat{a}, \hat{a}^\dagger] = 1$; the atomic field operator $\hat{\Psi}_j(x, t)$ destroys an atom in the internal state $|j = 1, 2\rangle$ at position $x$ and time $t$, the commutation relations are $[\hat{\Psi}_i(x, t), \hat{\Psi}_j^\dagger(x', t)]_\pm = \delta_{ij}\delta(x - x')$ (where $\pm$ indicate the anti- and commutation relations, depending on whether the atoms are fermions or bosons, respectively). Cavity and atomic operators commute at equal times. The Hamiltonian governing the dynamics can be decomposed into the sum of atoms, electromagnetic field, and atom-field interactions. The atomic Hamiltonian reads

$$\hat{\mathcal{H}}_A = \sum_{j=1,2} \int dx\, \hat{\Psi}_j^\dagger(x)\hat{H}_j(x)\hat{\Psi}_j(x) + 2U_{12}\int dx\, \hat{\Psi}_1^\dagger(x)\hat{\Psi}_2^\dagger(x)\hat{\Psi}_2(x)\hat{\Psi}_1(x), \tag{18}$$

where

$$\hat{H}_j(x) = -\frac{\hbar^2\partial_x^2}{2m} + V_{\text{cl}}^{(j)}\cos^2(kx) + \frac{1}{2}\int dx'\, U_{jj}(x - x')\hat{\Psi}_j^\dagger(x')\hat{\Psi}_j(x') - \delta_{j,2}\hbar\Delta_a|2\rangle\langle 2|. \tag{19}$$

Here, $\Delta_a = \omega_L - \omega_0$ is the detuning between the laser frequency $\omega_L$ and the atomic transition frequency in the frame rotating with the laser frequency $\omega_L$, $V_{\text{cl}}^{(j)}$ is the state-dependent depth of the optical lattice tightly binding atoms to its minima. The optical lattice potential is due

to light shifts by a laser driving a different atomic transition. The other parameters are the collision rate $U_{jj}$, which is also state dependent, and the collision rate $U_{12}$ between atoms in different electronic states. Since the laser is here described by a classical field, the only quantum field is the cavity mode at frequency $\omega_C = ck$, with $c$ the speed of light. The field Hamiltonian in the frame rotating with the laser frequency takes the form

$$\hat{\mathcal{H}}_C = -\hbar \Delta_C \hat{a}^\dagger \hat{a}, \tag{20}$$

where $\Delta_C = \omega_L - \omega_C$. Finally, the Hamiltonian describing the interaction between the atomic dipoles and the electric fields is given by

$$\hat{\mathcal{H}}_{\text{int}} = \hbar \int dx \hat{\Psi}_2^\dagger(x) \hat{\Psi}_1(x)(g(x)\hat{a} + \Omega_z(x)) + \text{H.c.}, \tag{21}$$

where $g(x) = g_0 \cos(kx)$ is the cavity vacuum Rabi frequency and $\Omega_z(x)$ is the position-dependent (real-valued) Rabi frequency of a laser propagating along the $z$ direction and coupling to the atomic transition (compare Fig. 1). 
  We now assume that the photon scattering processes are elastic, which is valid assuming that the largest frequency is the detuning $|\Delta_a|$. In particular, $|\Delta_a| \gg \gamma$, where $\gamma$ the radiative linewidth of the excited state, and $|\Delta_a| \gg \Omega, g_0\sqrt{n_{\text{cav}}}, |\Delta_C|$, namely, the detuning is much larger than the strength of the coupling between the ground and excited state, where $n_{\text{cav}} = \langle \hat{a}^\dagger \hat{a} \rangle$ is the mean intracavity photon number. In this regime we can eliminate the excited state approximating [27, 28]

$$\hat{\Psi}_2(x,t) \simeq \frac{g(x)}{\Delta_a} \hat{\Psi}_1(x,t) \hat{a}(t) + \frac{\Omega_z(x)}{\Delta_a} \hat{\Psi}_1(x,t), \tag{22}$$

which is valid to lowest order in the expansion in $1/|\Delta_a|$. Using Eq. (22) in the Heisenberg equation of motion for the field operator $\hat{\Psi}_1(x,t)$ results in the equation (we now drop the subscript from the field operator: $\hat{\Psi}_1 \equiv \hat{\Psi}$)

$$\dot{\hat{\Psi}} = -\frac{i}{\hbar}[\hat{\Psi}, \hat{\mathcal{H}}_A] - i\frac{\Omega_z(x)^2}{\Delta_a}\hat{\Psi} - iU_0(x)\,\hat{a}^\dagger\hat{\Psi}\hat{a} - iS(x)\left(\hat{a}^\dagger\hat{\Psi} + \hat{\Psi}\hat{a}\right), \tag{23}$$

where we have taken care to keep the ordering between cavity and atomic operators. This equation is now solely coupled to the Heisenberg-Langevin equations for the cavity field:

$$\dot{\hat{a}} = -\kappa\hat{a} + i\left(\Delta_C - \int dx U_0(x)\hat{n}(x)\right)\hat{a} - i\int dx S(x)\hat{n}(x) + \sqrt{2\kappa}\hat{a}_{\text{in}}, \tag{24}$$

which depends on the atomic operators through the atomic density

$$\hat{n}(x) = \hat{\Psi}^\dagger(x)\hat{\Psi}(x). \tag{25}$$

Equation (24) includes the quantum noise due to the cavity losses, with $\kappa$ the cavity loss rate and $\hat{a}_{\text{in}}(t)$ is the input noise operator, with $\langle \hat{a}_{\text{in}}(t)\rangle = 0$ and $\langle \hat{a}_{\text{in}}(t)\hat{a}_{\text{in}}^\dagger(t')\rangle = \delta(t-t')$ [84]. The other parameters are the frequency $U_0(x) = g(x)^2/\Delta_a$, which scales the depth of the intracavity potential generated by a single photon, and scattering amplitude $S(x) = g(x)\Omega_z(x)/\Delta_a$ [26, 27].
  We now eliminate the cavity degrees of freedom from the atomic dynamics by identifying the time-scale $\Delta t$ over which the atomic motion does not significantly evolve while the cavity field has relaxed to a state which depends on the atomic density at the given interval of time. This is verified when $\Delta t \gg T_c$, where $T_c = 1/|\Delta_C + i\kappa|$, and $\sqrt{\omega_R E_{\text{kin}}} \ll \hbar/\Delta t$, with $\omega_R = \hbar k^2/(2m)$ the recoil frequency and $E_{\text{kin}}$ the mean kinetic energy [36]. We also require

that the coupling strengths between atoms and fields, which determine the time scale of the evolution due to the mechanical effects of the interaction with the light, are much smaller than $1/\Delta t$. In this limit, we replace the field operator with its coarse-grained average $\hat{a}_{\mathrm{st}}(\bar{t})$, defined as

$$\int_{\bar{t}}^{\bar{t}+\Delta t} \hat{a}(\tau)d\tau/\Delta t = \hat{a}_{\mathrm{st}}(\bar{t}),$$

such that $\int_{\bar{t}}^{\bar{t}+\Delta t} \dot{\hat{a}}_{\mathrm{st}}(\tau)d\tau = 0$, with $\dot{\hat{a}}$ given in Eq. (24). The "stationary" cavity field is a function of the atomic operators at the same (coarse-grained) time, and in particular it takes the form

$$\hat{a}_{\mathrm{st}}(\bar{t}) = \frac{\int dx S(x)\hat{n}(x,\bar{t})}{(\Delta_C - \int dx U_0(x)\hat{n}(x,\bar{t})) + i\kappa} + \frac{i\sqrt{2\kappa}\hat{\bar{a}}_{\mathrm{in}}(\bar{t})}{(\Delta_C - \int dx U_0(x)\hat{n}(x,\bar{t})) + i\kappa}, \qquad (26)$$

with $\hat{\bar{a}}_{\mathrm{in}}$ the input noise averaged over $\Delta t$, such that in the coarse-grained time scale $\langle\hat{\bar{a}}_{\mathrm{in}}(\bar{t})\rangle = 0$ and $\langle\hat{\bar{a}}_{\mathrm{in}}(\bar{t})\hat{\bar{a}}_{\mathrm{in}}^\dagger(\bar{t}')\rangle = \delta(\bar{t} - \bar{t}')$. Note that the commutation relations of the new operators are modified, and in particular the commutator between $\hat{a}_{\mathrm{st}}$ and $\hat{a}_{\mathrm{st}}^\dagger$ scales as $T_c/\Delta t$, The quantum noise term is an effective term which provides the same averages as the corresponding quantum-noise operator one gets by formally integrating Eq. (24). It can be neglected when the mean intracavity photon number is larger than its fluctuations or by integrating for sufficiently long time. In this limit the dynamics is coherent and the field at the cavity output,

$$\hat{\bar{a}}_{\mathrm{out}}(t) = \sqrt{2\kappa}\hat{a}_{\mathrm{st}} - \hat{\bar{a}}_{\mathrm{in}}, \qquad (27)$$

allows one to monitoring the state of the atoms [25, 33, 84, 85].

Using Eq. (26) in place of the field $\hat{a}$ in Eq. (23), and discarding the noise term, leads to an equation of motion for the atomic field operator which depends solely on the atomic variables [27, 28]:

$$\dot{\hat{\Psi}} = -\frac{i}{\hbar}[\hat{\Psi}, \hat{\mathcal{H}}_A] - i\frac{\Omega_z(x)^2}{\Delta_a}\hat{\Psi} - i\frac{\int dx' S(x')\hat{n}(x')}{\Delta_C}S(x)\hat{\Psi}(x) - iS(x)\hat{\Psi}(x)\frac{\int dx' S(x')\hat{n}(x')}{\Delta_C}, \qquad (28)$$

where we have assumed $|\Delta_C| \gg |U_0|n, \kappa$ and thus discarded the terms scaling with $U_0$ in the denominators of Eq. (26) (see [28] for a discussion). The term of the right-hand side of Eq. (28) can be cast in terms of the commutator between $\hat{\Psi}$ and the effective Hamiltonian $H = H_A + H_{\mathrm{CQED}}$, where [28]:

$$H_{\mathrm{CQED}} = \hbar\int dx \frac{\Omega_z(x)^2}{\Delta_a}\hat{n}(x) + \frac{1}{\Delta_C}\hbar\left(\int dx\, S(x)\hat{n}(x)\right)^2, \qquad (29)$$

which contains an infinitely ranged density-density interaction. This interaction is attractive for $\Delta_C < 0$, which is the regime we assume in this work.

The Hubbard model is obtained when the external optical lattice is sufficiently deep to tightly bind the atoms in the lowest band and when the cavity interactions are a sufficiently small perturbation. We denote by $K$ the number of lattice sites and perform the Wannier decomposition of the atomic field operator $\hat{\Psi}(x) = \sum_i w_i(x)\hat{b}_i$ [86, 87]. Here, $w_i(x)$ denotes the Wannier function of the classical optical lattice that is centered at lattice sites $x_i = ia$, with $a = \lambda_0/2$ the lattice periodicity, while $\hat{b}_i$ annihilates a particle at the corresponding lattice site. We further assume the scaling $g(x) = \tilde{g}(x)/\sqrt{K}$ (thus $S(x) = \tilde{S}(x)/\sqrt{K}$), which is equivalent to assuming that the cavity mode volume scales linearly with the size of the lattice [28, 88]. This scaling is equivalent to Kac's scaling and warrants that the energy is extensive despite the

all-to-all interactions (even though it remains non-additive). The Hubbard term due to the cavity global interactions takes the form (recall $\Delta_C < 0$) [33, 34]:

$$-\frac{1}{|\Delta_C|}\left(\int dx\, S(x)\hat{n}(x)\right)^2 \approx$$
$$-\frac{1}{K|\Delta_C|}\sum_{i,j}\left(\int dx\,\tilde{S}(x)w_i(x)^2\right)\left(\int dx\,\tilde{S}(x)w_j(x)^2\right)\hat{n}_i\hat{n}_j, \qquad (30)$$

with $\hat{n}_i = \hat{b}_i^\dagger \hat{b}_i$ the occupation of site $i$. For $\tilde{S}(x) = S_0\cos(kx)$, corresponding to a homogeneous transverse pump $\Omega_z(x) = \Omega$, then $S_0 = \tilde{g}_0\Omega/\Delta_a$ and

$$\int dx\,\tilde{S}(x)w_i(x)^2 = S_0\int dx\cos(kx)w_i(x)^2 = (-1)^i S_0\left|\int dx\cos(kx)w_i(x)^2\right|. \qquad (31)$$

Thus, one obtains Eq. (5) with the coefficient

$$U_1 = \frac{S_0^2}{|\Delta_c|}\left(\int dx\cos(kx)w_i^2\right)^2, \qquad (32)$$

while the remaining terms in the tight binding approximation yield the Hubbard model. Let us note that the (coarse-grained) field $\hat{\bar{a}}_{\text{out}}(t)$ may be expressed as (neglecting the noise term in the coarse-grained limit)

$$\hat{\bar{a}}_{\text{out}}(t) = \frac{\sqrt{2\kappa}S_0}{\Delta_c\sqrt{K}}\sum_i(-1)^i\hat{n}_i - \hat{\bar{a}}_{\text{in}}. \qquad (33)$$

Hence the mean electric field at the cavity output is proportional to the expectation value of the population imbalance,

$$E_{\text{out}}(t) \propto \left\langle\sum_i(-1)^i\hat{n}_i\right\rangle = \langle I\rangle(t), \qquad (34)$$

where we used the definition in Eq. (6). We remark that, for the same reason that the time-scale separation allows one to derive a Hamiltonian dynamics for the atomic motion, thus discarding the commutation relation between the cavity field operators, then projection noise due to cavity leakage and photo-detection can be neglected because it is of the same order as the non-adiabatic corrections.

## B  Nonergodic regime for strong all-to-all interactions

Even in the absence of disorder strong long-range interactions may lead to nonergodic behavior manifesting itself e.g. in the logarithmic or sublinear growth of the entanglement entropy for a sudden quench [55, 89, 90]. This behavior seems to be distinct from MBL. An appearance of such a regime in the model studied here may be visualized considering mean gap ratio for large $U_1$ values as shown in Fig. 15. For large $U_1$ system tends towards gap ratio $\bar{r}$ corresponding to Poisson statistics.

Similar conclusions may be obtained from the time dynamics of the density-density correlation function $C(t)$ and of bipartite entanglement entropy $S(t)$ starting from initial random Fock state as shown in Fig. 16. Large $U_1$ leads to slower decay of correlations accompanied by a slower sublogarithmic even growth of the entanglement entropy. While for $K > 16$ we cannot follow the dynamics for sufficiently long times the results indicate a clear saturation of $C(t)$ for $U_1 = 25$ in agreement with mean gap ratio values.

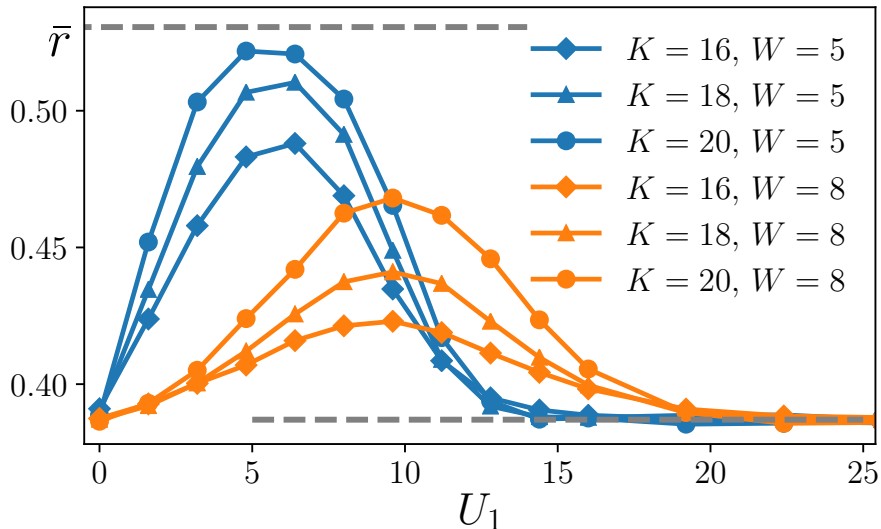

Figure 15: Mean gap ratio $\bar{r}$ as a function of all-to-all coupling $U_1$ for different system sizes and fixed disorder strength $W$. Observe that for a given disorder strength $W$, the mean gap ratio $\bar{r}$ first increases, consistently with an appearance of the ergodic phase shown in Fig. 2. Then, for large $U_1$, the gap ratio tends back towards the value corresponding to Poisson statistics indicating an occurrence of another non-ergodic phase. The mean gap ratio is determined in the center of the spectrum (for $\epsilon_n = (E_n - E_{\min})/(E_{\max} - E_{\min}) \approx 0.5$).

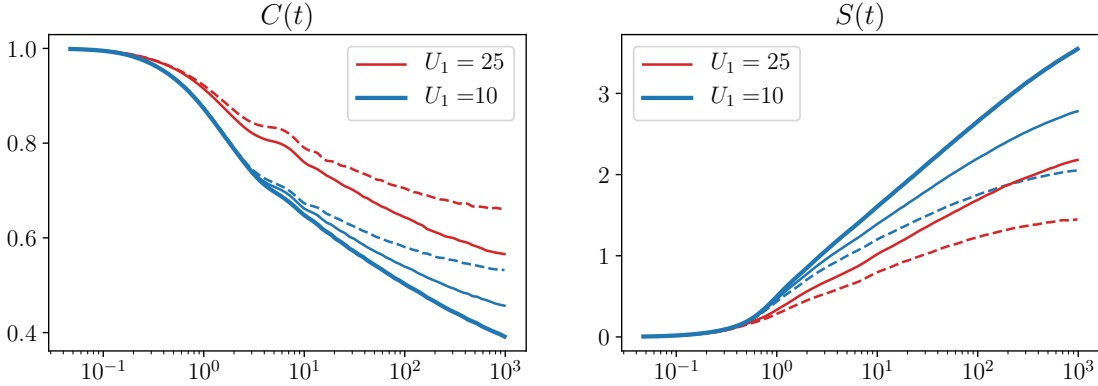

Figure 16: Time evolution of the correlation function, $C(t)$ (left) and of the entanglement entropy, $S(t)$ after a quench for $W = 8$ and two values of $U_1$, and the system initially prepared in a random Fock state. The dashed, narrow and broad line correspond to system size $K = 16, 18, 20$ respectively. Observe that large $U_1$ lead to slower entropy growth and slower decay of the correlation function, in both cases the dependence on the system size remains significant.

## C   Time evolution with Chebyshev expansion technique

Time evolution of fermionic systems at half filling with $K = 16$ (and smaller) can be obtained easily by full exact diagonalization of the Hamiltonian matrix followed by exact calculation of the evolution operator $U(t)$ for arbitrary time $t$.

To deal with larger system sizes we employ the expansion of the evolution operator into series involving Chebyshev polynomials [91, 92]

$$U(t) \approx e^{-ibt}\left(J_0(at) + 2\sum_{k=1}^{N}(-i)^k J_k(at)T_k(\mathcal{H})\right), \tag{35}$$

where $a = (E_{max} - E_{min})/2$, $b = (E_{max} + E_{min})/2$, the Hamiltonian is rescaled $\mathcal{H} = \frac{1}{a}(H - b)$ so that spectrum of $\mathcal{H}$ belongs to the $[-1, 1]$ interval, $J_k(t)$ is the Bessel function of the order $k$ and $T_k(x)$ is the Chebyshev polynomial of order $k$. The number of terms $N$ needed to assure convergence of the expansion (35) to time $t_{max}$ is $N \approx 2at_{max}$ [93].

The time-evolution of the initial state $|\psi_0\rangle$ is given by

$$|\psi(t)\rangle \approx e^{-ibt}\left(J_0(at)|\psi_0\rangle + 2\sum_{k=1}^{N}(-i)^k J_k(at)T_k(\mathcal{H})|\psi_0\rangle\right) \tag{36}$$

and reduces to matrix-vector multiplications

$$T_k(\mathcal{H})|\psi_0\rangle = 2\mathcal{H}T_{k-1}(\mathcal{H})|\psi_0\rangle - T_{k-2}(\mathcal{H})|\psi_0\rangle, \tag{37}$$

where the recursion relation satisfied by Chebyshev polynomials was used. In order to get $|\psi(t)\rangle$ we generate iteratively a sequence of $N$ vectors $|\psi_0\rangle, T_1|\psi_0\rangle, ..., T_N|\psi_0\rangle$. To reach long times of time evolution $t_{max} \approx 10^3$ one needs relatively large $N$ which increases memory consumption. Therefore we split the time interval $[0, t_{max}]$ into parts $[0, \Delta t], [\Delta t, 2\Delta t], ...$ in such a way that $|\psi((n+1)\Delta t)\rangle$ can be calculated from the state $|\psi(n\Delta t)\rangle$ with the expansion (36) involving only a limited number of terms e.g. $-N \approx 1000$ which allows us to obtain time evolution for the system size $K = 20$ with memory consumption smaller than 5GB (performing the matrix-vector multiplications in PETSc).

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
