# Peer review of "Many-body localization in presence of cavity mediated long-range interactions"

_SciPost Physics, doi:SciPost Phys. 7, 008 (2019)_

## Round 2 · Referee Report · Anonymous (Referee 1) · 2019-2-14

Strengths

1- the use of cavity QED to study MBL physics is very interesting, since the CQED could prepare fancy interaction form but not ground state, and the MBL physics explores properties not only about ground state. The detection method is very novel. 2- the phenomenological scaling effect produces very convincing results in Fig.3.

Weaknesses

1- the effects of cavity is not only providing this long-range interaction. The quantum optical properties such as photon induced contrast lost are not discussed. These properties, however, should be a very significant limitation for optical-related detecting method as suggested by authors. 2- authors should cite more related articles such as those experiments using cavity-feedback interaction. Examples include but not limited to [1-2] for Boson and [3] for Fermion.

[1] M.H. Schleier-Smith, I.D. Leroux, and V. Vuletić, Phys. Rev. A 81, 021804(R) (2010) [2] E. Davis, G. Bentsen, L. Homeier, T. Li, and M. Schleier-Smith. "Photon-mediated spin-exchange dynamics of spin-1 atoms," Phys. Rev. Lett. 122, 010405 (2019) [3] Boris Braverman, et al. "Near-Unitary Spin Squeezing in 171Yb." arXiv preprint arXiv:1901.10499 (2019)

Report

The proposal to use cavity feedback long-range interaction to study MBL physics under Hubbard Hamiltonian is both physically interesting and feasible. The long-range all-to-all interaction provides an experimental platform for unexplored MBL phases, and the detection method is also innovative. The paper is already in a good shape despite few places needed to be improved as I list in the requested changes.

The system contains a lot of complicated parameters, such as all-to-all energy scale, on-site energy scale, disorder scale, etc. In simplified system with random all-to-all and disorder, according to [1], sec III.C.1, an MBL phase is impossible in the infinite range of interactions setup. In this work, however, the authors observed this MBL behavior with strong long-range interaction, which is quite interesting. Though, it should be careful that whether this is MBL or MBL spin glass phase.

However, the explanation for eigenstate statistics is not enough to show the MBL, and the authors enhance the argument by providing dynamical investigations. The ergodicity breaking observed proves the MBL phase.

In general this is a good paper and I recommend for minor revision after answering and solving the weaknesses and requested changes.

[1] Dmitry A. Abanin, Ehud Altman, Immanuel Bloch, Maksym Serbyn, "Many-body localization, thermalization, and entanglement" arXiv preprint arXiv:1804.11065 (2018)

Requested changes

1- should cite more related experimental articles for this cavity QED system, especially those with optical lattice (say, arXiv:1901.10499). 2- should discuss about the mean gap ratio's fluctuation. In Fig.2, the range spans only from 0.40 to 0.52. The necessary uncertainty analysis is needed to determine whether this phase diagram has enough signal-to-noise ratio. 3- the scaling exponents, in the main text, is $\nu(U=1)=1, \nu(U=4)=1.5$. However in Fig.3, they're $1.3, 1.8$ respectively. I believe this is a typo that should be fixed. 4- discuss possibilities that this is a MBL spin glass phase. Not necessary but if possible, the authors should try to distinguish these by using the spin glass order as introduced in [1].

[1] Jonas A. Kjäll, Jens H. Bardarson, Frank Pollmann, "Many-body localization in a disordered quantum Ising chain" Phys. Rev. Lett. 113, 107204 (2014)

---

## Round 2 · Referee Report · Anonymous (Referee 3) · 2019-3-14

Strengths

  • Timeliness and experimental relevance

Weaknesses

  • Technicality
  • insufficient discussion of the continuous measurement of the scattered light

Report

This paper presents a numerical study of Many-Body Localization (MBL) in an extended one dimensional Hubbard model. The main originality is the inclusion of infinite-range interactions, motivated by the experimental progresses with quantum gases in optical cavities, where such an interaction has been demonstrated.

The core of the paper is dedicated to the study of the consequences of this particular interaction on the MBL phase, showing that (i) it shifts the transition to higher disorder strength and (ii) the MBL phase still exists even for moderate strength for this interaction. The observables used are the long time dynamics of density correlations and the growth of entanglement entropy, which are the canonical tools for characterizing MBL.

The paper is timely, since it deals with one of the most active research topic in quantum gases and quantum physics in general. It proposes the combination of two experimentally accessible configurations, the production of disorder and controlled interactions, and the use of a high finesse cavity. While no experimental setup exists so far that combines all of these, it is reasonable to expect that a new generation of experiments could have the capabilities to realize the proposal, and the present paper may be motivation for such an experimental development. I appreciate the efforts done the authors to treat the cases closest to the experiments, in particular the quasi-random lattice case.

As an experimentalist, I cannot judge the validity of the numerical methods or approximations used, but I find the conclusions on the persistence of MBL as well as the extra delocalization induced by the infinite range interaction convincing (in particular in figure 6).

The aspect I found most interesting is the existence of a new observable, the cavity field leaking from the cavity, as a way to further characterize the MBL state and transition. I am however quite disappointed by the treatment of this part. The premise that the measurement is non-destructive is obviously wrong: the operator $I(t)$ does not commute with the Hamiltonian. Therefore, the measurement will induce a new dynamics, which the authors miss. A very important question arises: does the MBL phase indeed survives the continuous measurement process, or does the measurement brings in new states ? I understand that the theoretical treatment of such an open system would be much more involved and probably beyond the scope of the study, but without any discussion of this aspect, I am not convinced that indeed the cavity field leakage can indeed be interpreted as indicating anything regarding MBL.

I would recommend that the authors discuss this point, and leave out for a possible further study the exact interpretation of the cavity leakage. I also think that the material provided by the authors in the other sections is enough to support the main claims of the paper. Provided that the technical aspects of the paper are confirmed by a theoretician, I can propose publication in Scipost when the following changes are made.

Requested changes

  1. figure 1: \kappa and \Omega_z are introduced but nowhere used in the text nor in the discussion. I have the feeling that the processes yielding the long range interaction (photon exchanges with the cavity field) could be better described, even though this has been covered by the literature. I suggest to add an appendix with a summary of the derivation.
  2. Eq (6): it is not clear over which ensemble the minimum and maximum is supposed to be taken. Is it over a range of energies, over disorder realizations ? Please make this definition more explicit.
  3. page 4, last paragraph: the calculation is done for exact commensurability, which is a very special case strongly favoring super radiance. Is there any reason to think that this is representative of the more general case ? What should one qualitatively expect in other situations ? This is of interest since any experimental realization will have inhomogeneities in the filling.
  4. page 5: please define the acronyms PETSc/SLEPs or remove them from the text.
  5. page 6: missing ’s’ in the sentence ‘’So far…. indicates…’’
  6. page 6, last paragraph of section 3: this is quite confusing: which non ergodicity are the authors talking about ? I suggest to enhance this paragraph in order to include a short physical discussion of this other regime at large interactions, in particular to convince the reader that it cannot be favored by disorder and explain the observations.
  7. page 7 last sentence of the first paragraph: typo ‘regime where’
  8. page 6 last sentence of the page: I assume that the lattice is split in two subsystems separated in space. Considering the peculiar interaction pattern, I have a naive question: would it make sense to split the system in even/odd sites ? That would not have the interpretation of localization in space, but could help representing the memory of the initial state.
  9. page 8, figure 4: the persistence of size effects in the entanglement entropy for U_1=1 while the density correlations do not depend on size could simply to suggest that there are other degrees of freedom (e.g. phases) which contribute to entanglement. Please comment.
  10. figure 7: letters designating the sub panels should be mentioned in the caption.
  11. page 13: the measurement is not ‘non destructive’. Please comment on this point (see above).
  12. page13, Eq 14: this quantity is measured in principle for one given realization of the disorder (and of the experiment). Here, it is not clear whether the authors consider the average (and which average). Please comment if this is still to be used.
  13. Figure 9: The superposition of both C(t) and I^2(t) on the same graph brings more confusion than clarity. Is the scale for the two the same ?
  14. page 13-14: its seems that figure 9a shows that even in the ergodic phase, I^2(t) can reach a non zero steady state. I agree that this most likely means that the state stays close to the super radiant phase, but is casts doubts on the possibility to use I^2(t) as unambiguously indicating ergodicity. In general, I find the interpretation of these results confusing.
  15. page14, last paragraph: an off-resonant laser with random intensity would not be quasi-random but truly random (as is typical for speckle patterns. see for exemple the work of the Aspect group).

---

## Round 2 · Referee Report · Anonymous (Referee 2) · 2019-3-14

Strengths

1- Experimental relevance of the results.

Weaknesses

I have two main concerns outlined in the report.

Report

1) Is the saturation of $S_P$ sufficient to claim MBL, in particular since the $S_C$ is growing linearly at long-times (assuming the thermodynamic limit)? How are these contributions behaving in systems with long-range interactions? Can the authors observe a transition between the discussed phases and the phase with dominant long-range interactions?

2) In systems with long-range interactions, the convergence of results with system size is typically very slow. Can the effects of infinite-range interactions be distinguished from the ones obtained by finite range interactions over several sites for systems with 20 sites? Several other approaches can be used to calculate the time evolution for larger systems. For example, using the time-dependent variational principle, or by transforming the model into a short range model with a self-consistent population imbalance which enables the use of standard methods for simulations of 1D short-range systems.

Besides above it would be interesting to see if there is any qualitative difference considering U1=8 in figures 4 and 5 which is in the ergodic regime for the considered system size and disorder strength?

Requested changes

  • Use a consistent reference to sub figures. Sometimes left/right is used and sometimes a),b).

---

## Round 3 · Referee Report · Anonymous (Referee 2) · 2019-6-12

Report

The authors sufficiently address raised questions/concerns, hence I recommend the work for publishing.

---

## Round 3 · Author Response

Dear Editor,

We would like to resubmit our manuscript "Many-body localization in presence of cavity mediated long-range interactions ". Following the Referees' recommendations and comments, we have substantially reworked and expanded the manuscript. We provided additional details in Appendixes and modified the paper, in particular the part related to the cavity output. We believe that the new version takes into account all recommendations and remarks of the Referees. Our detailed responses to all comments by the Referees and the list of changes made are appended below.

We hope that the present manuscript meets the standards of Scipost Physics publication.

With best regards,
The authors

(Fragments of reports of referees in quotation marks)

Reply to Referee 1

Weaknesses

Referee: "1- the effects of cavity is not only providing this long-range interaction. The quantum optical properties such as photon induced contrast lost are not discussed. These properties, however, should be a very significant limitation for optical-related detecting method as suggested by authors."

We thank the referee for pointing this out. Indeed, in the regime we consider shot noise and the effect of projection noise on the atomic dynamics are negligible. We have added an extensive appendix which summarizes the derivation of our Hubbard model, the regime of its validity, as well as the reference to the previous literature, where the effective model has been derived. Below we further discuss this point in reply to the specific questions of the third referee.

Referee: "2- authors should cite more related articles such as those experiments using cavity-feedback interaction. Examples include but not limited to [1-2] for Boson and [3] for Fermion."

[1] M.H. Schleier-Smith, I.D. Leroux, and V. Vuletic, Phys. Rev. A 81, 021804(R) (2010)

[2] E. Davis, G. Bentsen, L. Homeier, T. Li, and M. Schleier-Smith. "Photon-mediated spin-exchange dynamics of spin-1 atoms," Phys. Rev. Lett. 122, 010405 (2019)

[3] Boris Braverman, et al. "Near-Unitary Spin Squeezing in 171Yb." arXiv preprint arXiv:1901.10499 (2019)

The experimental works on many-body cavity quantum electrodynamics are numerous and it is impossible to mention them all in an original paper. We have majorly focused on optomechanical dynamics, where the external degrees of freedom directly couple with the optical field. We have nevertheless added a PRL related to [1] (as [1] is theoretical), and further recent works from various groups on optomechanical dynamics - new references [39-42].

Requested changes:

Referee: "1- should cite more related experimental articles for this cavity QED system, especially those with optical lattice (say, arXiv:1901.10499)."

We have implemented the referee's request, at least partially. See comment above.

Referee: "2- should discuss about the mean gap ratio's fluctuation. In Fig.2, the range spans only from 0.40 to 0.52. The necessary uncertainty analysis is needed to determine whether this phase diagram has enough signal-to-noise ratio."

Actually the range spans from 0.39 to 0.53 i.e. the relevant range between Poisson and GOE predictions. In a modified figure we made it clearer by changing the color bar labels. The error of each data point is smaller than the symbol size as mentioned already in the caption of Fig.3.

Referee: "3- the scaling exponents, in the main text, is}$ $\nu$(U=1)=1,$\nu$\textit{(U=4)=1.5. However in Fig.3, they're 1.3,1.8 respectively. I believe this is a typo that should be fixed."

It has been fixed. Thank you for noting that.

Referee: "4- discuss possibilities that this is a MBL spin glass phase. Not necessary but if possible, the authors should try to distinguish these by using the spin glass order as introduced in [1]."

The different phases: paramagnetic and spin-glass, are identified in [1] for the Ising model. They are due to disorder preserved order manifesting the transition between paramagnetic and ferromagnetic phases of the ground state of Ising model. For XXZ model there is no analog of that -- our model generalizes XXZ to all-to-all interaction case. The aim of our work is the identification of possible MBL phase for the cavity-based model, study of additional possible phases within this MBL would have to, necessarily, involve a similar study for a celebrated, short-range XXZ model too. That would expand our work beyond proportions and put the emphasis on different aspects. So while grateful for the suggestion, we leave this aspect to a future work.

Reply to Referee 2

Referee: "1) Is the saturation of SP sufficient to claim MBL, in particular since the SC is growing linearly at long-times (assuming the thermodynamic limit)? How are these contributions behaving in systems with long-range interactions? Can the authors observe a transition between the discussed phases and the phase with dominant long-range interactions?"

Saturation of $S_P$ is sufficient provided it is at a low value, much lower than for the delocalized case. Saturation of $S_P$ indicates that the redistribution of particles has been realized already; the remaining growth of the entropy is of ``configuration'' type. While we observe that $S_C$ shows linear growth for small $U_1$ pointing towards delocalization this growth saturates at the system dependent value for small $U_1$ (Fig.7). This is interpreted as a mixing of states in time within the submanifold of closely spaced in energy states for Anderson case, $U=0$ or for MBL , $U=1$ panel b). This growth saturates (faster for slightly bigger $U_1$) at still very low entropy values. It is true that this saturation level diverges probably in the thermodynamic limit, but similarly logarithmically growing entropy in standard MBL diverges in this limit at long times.

The growth of $S_C$ with system size is associated with a growing number of configurations in two subsystems. Analysis of very low $U_1$ values allows us to identify two phenomena - redistribution of particles (due to $S_P$) and entanglement of different configurations. In the presence of density-density all-to-all interactions, as in the case of the system studied, the standard definition of MBL in the language of LIOMs as given by (11) and (12) breaks down. Proper understanding of the system requires long range coupling between LIOMs (13) which is responsible for growth of $S_C$.

One could argue that, therefore, MBL is destroyed. In the spirit of [11] we rather classify the observed behavior as a nonstandard MBL.

Referee: "2) In systems with long-range interactions, the convergence of results with system size is typically very slow. Can the effects of infinite-range interactions be distinguished from the ones obtained by finite range interactions over several sites for systems with 20 sites? Several other approaches can be used to calculate the time evolution for larger systems. For example, using the time-dependent variational principle, or by transforming the model into a short range model with a self-consistent population imbalance which enables the use of standard methods for simulations of 1D short-range systems."

The Referee is obviously right about the significance of finite size effects that cannot be avoided for all-to-all interactions. To minimize their influence we used in calculations periodic boundary conditions while we admit that the cavity-model requires open boundary conditions. Going to TDVP for larger system sizes forms a large independent study which we have commenced and we hope will form a basis of our future analysis. We thank the referee for the suggestion of `` transforming the model into a short range model with a self-consistent population imbalance''. If we understood that correctly the idea is to decorrelate the interaction term $\sum_{i,j}(-1)^{i+j}n_i n_j$ as $2I(t)\sum_j(-1)^{j}n_j - \left(I(t)\right)^2$. This transforms the long-range interaction problem in a quenched disorder to short-range interactions with disorder having a time-dependent disorder proportional to a time-dependent imbalance. Such a time-dependent disorder might destabilize localization. We have made such an approximation and our numerical studies indicate that the opposite is true, the disorder, due to its specific shape, we believe, stabilizes slightly non-ergodic localized behavior.

Referee: "Besides above it would be interesting to see if there is any qualitative difference considering U1=8 in figures 4 and 5 which is in the ergodic regime for the considered system size and disorder strength?"

We show the additional curve for $U_1=10$ in the Appendix B in discussion of large $U_1$ nonergodic regime (instead of suggested $U_1=8$ we take $U_1=10$ because for this value of all-to-all coupling the $\overline r$ is largest for $W=8$). It shows qualitatively the same behaviour as $U_1=3,5$ as far as the system size is concerned. The qualitative difference appears in Fig.4 between $U_1=1$ and other values -- for $U_1=1$ correlation curves $C(t)$ for different system sizes coincide pointing towards MBL in the thermodynamic limit. This cannot be concluded for $U_1=3,5,8$. Data for $U_1=3,5$ seem localized for the smallest size $K=16$ as a tendency for saturation in $C(t)$ is apparent (observe the tendency to flattening in dashed curves). This is not sen in $U_1=10$ curve being, even for $K=16$ in the delocalized regime.

Reply to Referee 3

Referee: "The aspect I found most interesting is the existence of a new observable, the cavity field leaking from the cavity, as a way to further characterize the MBL state and transition. I am however quite disappointed by the treatment of this part. The premise that the measurement is non-destructive is obviously wrong: the operator I(t) does not commute with the Hamiltonian. Therefore, the measurement will induce a new dynamics, which the authors miss. A very important question arises: does the MBL phase indeed survives the continuous measurement process, or does the measurement brings in new states ? I understand that the theoretical treatment of such an open system would be much more involved and probably beyond the scope of the study, but without any discussion of this aspect, I am not convinced that indeed the cavity field leakage can indeed be interpreted as indicating anything regarding MBL."

The effective Hamiltonian model considered in our work {is derived assuming a time-scale separation: the cavity field evolves much faster than the atomic motion. In this limit the dynamics of the atoms is determined in a coarse graining time step, over which cavity shot-noise is averaged out and projection noise is negligible. This derivation has been discussed in the literature over the years and it holds for the experimental regime of the ETH group.

In order to clarify this point we have included Appendix A in the present version, where the derivation of the model and the basic assumptions are reported, and we have clarified it further in the main text, when we discuss the measurement of the light at the cavity output. We note that monitoring of the population imbalance (in the sense of the coarse graining here discussed) has been implemented in several experiments, for instance it is used to reveal the Dicke phase transition in Baumann et al, Nature 2010, as well as the out-of-equilibrium dynamics in Hruby et al, PNAS 2018.}

Referee: "1. figure 1: $\kappa$ and $\Omega_z$ are introduced but nowhere used in the text nor in the discussion. I have the feeling that the processes yielding the long range interaction (photon exchanges with the cavity field) could be better described, even though this has been covered by the literature. I suggest to add an appendix with a summary of the derivation."

The appropriate Appendix A with the summary of the derivation of the effective Hamiltonian has been included. We thank the referee for this suggestion.

Referee: "2. Eq (6): it is not clear over which ensemble the minimum and maximum is supposed to be taken. Is it over a range of energies, over disorder realizations ? Please make this definition more explicit."

The answer is too simple: $\min(x,y)$ appearing in the definition of $r_n$ is a minimum of two entries $x$ and $y$. Gap ratio $r_n$ is then averaged over part of spectrum of the system and then over disorder realizations, as detailed in the text.

Referee: "3. page 4, last paragraph: the calculation is done for exact commensurability, which is a very special case strongly favoring super radiance. Is there any reason to think that this is representative of the more general case ? What should one qualitatively expect in other situations ? This is of interest since any experimental realization will have inhomogeneities in the filling."

MBL is a robust phenomenon observed for a variety of systems. There is no reason to believe that half filling is special (except that it corresponds to the biggest Hilbert subspace for a given number of sites $K$). Experimental evidence of MBL with fermions (without cavity) e.g. in I. Bloch group have also inhomogeneities in the filling. This makes a big difference as compared to ground state properties where in disorder-free case e.g. Mott insulator phase for bosons requires an integer filling.

Referee: "4. page 5: please define the acronyms PETSc/SLEPs or remove them from the text."

Done.

Referee: 5. page 6: missing ’s’ in the sentence ‘’So far…. indicates…’’

Corrected.

Referee: "6. page 6, last paragraph of section 3: this is quite confusing: which non ergodicity are the authors talking about ? I suggest to enhance this paragraph in order to include a short physical discussion of this other regime at large interactions, in particular to convince the reader that it cannot be favored by disorder and explain the observations."

Not to break the main line of the article we have chosen to present such a discussion in Appendix B.

Referee: "7. page 7 last sentence of the first paragraph: typo ‘regime where’ "

Corrected.

Referee: "8. page 6 last sentence of the page: I assume that the lattice is split in two subsystems separated in space. Considering the peculiar interaction pattern, I have a naive question: would it make sense to split the system in even/odd sites ? That would not have the interpretation of localization in space, but could help representing the memory of the initial state."

This is an interesting idea. We have calculated the entanglement entropy time evolution assuming the suggested splitting. The entropy behavior is very similar, such a splitting makes the growth slightly faster as the ``contact'' between splitted parts dramatically increases. No qualitative difference with respect to standard splitting is observed.

Referee: "9. page 8, figure 4: the persistence of size effects in the entanglement entropy for $U_1=1$ while the density correlations do not depend on size could simply to suggest that there are other degrees of freedom (e.g. phases) which contribute to entanglement. Please comment."

This is a good point which we hope we answer later in the text discussing in detail the growth of the configuration entropy $S_C$. The redistribution of particles among sites happens at earlier times and thus $C(t)$ for different sizes coincides. This point is discussed in detail later in the text.

Referee: "10. figure 7: letters designating the sub panels should be mentioned in the caption."

Done.

Referee: "11. page 13: the measurement is not "non destructive". Please comment on this point (see above)."

We have commented on this point in the reply to Referee 1 and at the beginning of the reply to this Referee. The manuscript has been modifed and extented to address explicitly this point, see for instance new Appendix A.

Referee: "12. page13, Eq 14: this quantity is measured in principle for one given realization of the disorder (and of the experiment). Here, it is not clear whether the authors consider the average (and which average). Please comment if this is still to be used."

While the experiment may be envisioned for a single realization of the disorder, the results presented are averaged over the disorder realizations. The average in Eq.~(15) p.13 $\langle I\rangle$ is the quantum-mechanical average of the observable $I$ averaged over different disorder realizations.

Referee: "13. Figure 9: The superposition of both C(t) and $I^2(t)$ on the same graph brings more confusion than clarity. Is the scale for the two the same ?"

The section about cavity output has been rewritten, in particular the imbalance may be measured as cavity field amplitude rather than as light intensity. This error in the previous version of the manuscript was corrected. That affects also the results and the figures. Following the suggestion we plot the correlation and the imbalance in separate panels.

Referee: "14. page 13-14: its seems that figure 9a shows that even in the ergodic phase,$I^2(t)$ can reach a non zero steady state. I agree that this most likely means that the state stays close to the super radiant phase, but is casts doubts on the possibility to use $I^2(t)$ as unambiguously indicating ergodicity. In general, I find the interpretation of these results confusing."

Unfortunately, large cavity output cannot be identified unambiguously with MBL character - it is proportional to the imbalance. Contrary to systems with short-range interactions discussed earlier, large imbalance occurs for the cavity model studied also in the situation when the system reorganizes itself, close to the ground state, in a density-wave-like pattern as experimentally verified in Esslinger group experiments for disorder free situation. Only in the regime of large disorder the cavity output may be correlated with the existence of MBL. We hope to make it more clear in the modified version of the paper. Moreover, we have also extended our calculations for the initial state $|11001100\ldots\rangle$. The measurement of light intensity at the cavity output provides more conclusive results about ergodicity of the system as we discuss in the modified manuscript.

Referee: "15. page14, last paragraph: an off-resonant laser with random intensity would not be quasi-random but truly random (as is typical for speckle patterns. see for example the work of the Aspect group)."

Indeed, thank you for this comment. We have reworded the first paragraph of this section accordingly.

---

## Round 3 · List of Changes

• Two appendixes added, first considers the derivation of the effective Hamiltonian, second strong all-to-all couplings case.
  • The signal measured as an output from the cavity is the field itself instead of intensity, description and figures are modified accordingly.
  • Typos have been corrected, a number of clarifications and additional explanations added to the manuscript.

---

## Editorial Decision

published